# Identification of key macrophage-related genes in systemic sclerosis–associated interstitial lung disease based on single-cell and bulk transcriptomic data

Ting Zhao, Yulin Wang, Fu-an Lin iD *

Department of Rheumatology and Immunology, Zhangzhou Affiliated Hospital of Fujian Medical University, Zhangzhou, Fujian Province, China

* Linfuan0084@126.com

## Abstract

### Background

Systemic sclerosis–associated interstitial lung disease (SSc-ILD) is a major clinical challenge with no effective treatments. It is also the leading cause of death in patients with systemic sclerosis. Thus, understanding its underlying molecular mechanisms, particularly those related to macrophage-related gene functions, is critical to address this urgent medical need.

### Methods

In this study, single-cell and transcriptomic data retrieved from a public database were analyzed to investigate the underlying molecular mechanisms of SSc-ILD. A series of comprehensive analyses was conducted, including cell–cell communication analysis, pseudotime trajectory analysis, and high-dimensional weighted gene co-expression network analysis, to identify pertinent genes linked to macrophage modules. Candidate genes were determined by intersecting differentially expressed genes (DEGs) with macrophage module genes. Subsequently, key genes were identified through protein–protein interaction (PPI) network analysis and gene expression validation. Various analytical procedures were used to evaluate the function of the key genes in the regulatory roles of SSc-ILD, including enrichment analysis, immune infiltration analysis, drug prediction, and molecular docking.

### Results

Of the 1515 DEGs and 400 macrophage module genes intersected, 50 candidate genes were identified. In particular, *ARG2*, *ELF3*, and *NKX2–1* emerged as key genes through subsequent PPI network analyses and gene expression evaluations. Enrichment analyses revealed a notable co-enrichment of the lysosomal pathway

**Data availability statement:** The datasets analysed during the current study are available in the Gene Expression Omnibus (GEO) database (GSE81292, GSE122960and GSE76808, http://www.ncbi.nlm.nih.gov/geo/).

**Funding:** This research was funded by Zhangzhou Affiliated Hospital of Fujian Medical University Doctoral Studio, Grant number: PDA202207. The funders had no role in study design, data collection and analysis, decision to publish, or preparation of the manuscript.

**Competing interests:** The authors declare no conflicts of interest.

with these key genes. Moreover, immune infiltration analysis revealed a strong negative correlation between NKX2–1 and monocytes, whereas ELF3 and ARG2 exhibited a positive association with activated dendritic cells. The molecular docking results showed that the binding energies of ARG2-SKA-111/cyclophosphamide and ELF3–voruciclib/cyclophosphamide were less than − 5 kcal/mol.

## Conclusion

The findings of this study highlight the key roles of ARG2, ELF3, and NKX2−1 in macrophage-related mechanisms of SSc-ILD, providing insights into potential therapeutic targets. Further research is necessary to explore their functional implications in disease progression and treatment.

---

## 1. Introduction

Systemic sclerosis (SSc) is a rare autoimmune disease, which is characterized by skin and organ fibrosis and vascular injury [1,2]. Interstitial lung disease (ILD) is a common clinical manifestation of SSc. Studies have shown that 74% − 85.5% of patients with SSc have ILD [3,4]. The etiology of SSc-ILD is complex, and its pathogenesis involves inflammation, alveolar epithelial cell injury, and pulmonary interstitial fibrosis [4]. The current treatment for SSc-ILD primarily involves corticosteroids combined with immunosuppressants; however, this approach can only partially control inflammation and delay disease progression. In addition, this treatment method has limited efficacy against vascular lesions and fibrosis, resulting in low clinical response rates [5,6]. Therefore, finding the key genes of SSc-ILD and providing new strategies for diagnosing and treating SSc-ILD is of great importance.

Macrophages play a crucial role within the innate immune system, and they are extensively found throughout various tissues in the body [7]. They phagocytize pathogens, remove cell debris, regulate inflammatory responses, and maintain tissue homeostasis [8]. Inflammation and fibrosis are the core pathological features in SSc-ILD [9]. Macrophages activate fibroblasts to differentiate into myofibroblasts and promote extracellular matrix (ECM) deposition by secreting profibrotic factors, such as transforming growth factor-beta (TGF-β) and interleukin-13 (IL-13) [10–12]. In patients with nonspecific interstitial pneumonia, macrophage-related gene clusters, such as collagen synthesis pathways and interferon (IFN) regulatory genes, are significantly associated with the progressive deterioration of lung function [13]. Therefore, macrophage-related genes may play an important role in developing SSc-ILD, but their specific molecular mechanism remains unclear.

Single-cell RNA sequencing (scRNA-seq) technology overcomes the limitations of traditional batch sequencing by analyzing transcriptomes at the single-cell level. It can accurately identify cell subsets and differences in functional status in tissues. In a study on pulmonary fibrosis, scRNA-seq revealed the heterogeneity of macrophage subsets such as SPP1hi and FABP4hi, as well as the significant differences in IFN signaling pathways between idiopathic pulmonary fibrosis (IPF) and SSc-ILD: IFN-γ

signaling in IPF was upregulated in macrophages and T cells, whereas the type I IFN pathway was activated in SSc-ILD [14]. By integrating scRNA-seq and chromatin accessibility analysis (scATAC-seq), it has been found that the epigenetic regulation of SPP1 + macrophages in SSc-ILD is abnormal, and their differentiation is driven by transcription factors such as the microphthalmia-associated transcription factor family and activator protein 1 family [15]. These factors regulate the expression level of profibrotic genes by opening chromatin regions.

Based on the single-cell data of SSc-ILD retrieved from a public database, our current study investigated the key cells in SSc-ILD. It also explored the cell communication and pseudo-timing analysis of key cell macrophages. In addition, the module genes in macrophages were screened through high-dimensional weighted gene co-expression network analyses (hdWGCNA), and differentially expressed genes (DEGs) in SSc-ILD were identified on the basis of transcriptome data. Subsequently, DEGs and module genes were crossed to obtain candidate genes. Key genes were obtained by PPI network analysis and expression verification. Enrichment analysis and immune infiltration analysis of key genes were also carried out. This study provides a new reference for the diagnosis and treatment of patients with SSc-ILD based on single-cell and transcriptome data combined with macrophage-related genes.

## 2. Materials and methods

### 2.1. Data collection

Data were obtained from the Gene Expression Omnibus database (https://www.ncbi.nlm.nih.gov/geo/). The GSE81292 dataset (sequencing platform: GPL18991), as the training set, included 15 SSc-ILD lung tissue samples and five normal lung tissue samples. The GSE76808 dataset (sequencing platform: GPL571), as the validation set, included 14 SSc-ILD lung tissue samples and four normal lung tissue samples. In addition, the single-cell dataset GSE122960 (sequencing platform: GPL20301) included two SSc-ILD lung tissue samples and eight normal lung tissue samples.

### 2.2. Treatment of scRNA-seq data

During the processing of scRNA-seq data from the GSE122960 dataset, quality control was initially performed using the Seurat package (v5.1.0) [16]. Cells meeting the criteria of $200 < \text{nFeature\_RNA} < 5000$, percent.mt < 10%, and $500 < \text{nCount\_RNA} < 5000$ were retained to remove low-quality samples. Then, the filtered data were normalized using the LogNormalize function with a scale factor of 10,000 to mitigate technical variation. Subsequently, the FindVariableFeatures function from the Seurat package (v 5.1.0) was employed to identify the top 2000 highly variable genes (HVGs) based on the coefficient of variation across cells, and the 10 most variable HVGs were labeled. Principal component analysis was conducted on these 2000 HVGs using the ScaleData function from the Seurat package (v 5.1.0). The first 20 principal components exhibiting statistical significance ($P < 0.05$) were selected for downstream analysis. Unsupervised clustering of all cells was performed using the Seurat package (v 5.1.0) with the resolution parameter set to 1. Finally, uniform manifold approximation and projection (UMAP) was run using default parameters for the number of neighbors and the minimum distance, based on the first 20 Harmony-corrected dimensions, to visualize the clustering results in a reduced-dimensional space. Finally, the marker genes provided in the literature [17,18] were applied to annotate the cell types of the different clusters. The annotation results were visualized in a UMAP plot, and the expression level of marker genes in different cell types was shown. Then, macrophages were selected as the key cells for subsequent analysis.

### 2.3. Identification of key cell–related modular genes

In obtaining genes associated with the key cells of SSc-ILD for the GSE122960 dataset, the hdWGCNA package (v 0.4.0) [19] was applied to construct a scale-free network. hdWGCNA is an R package for performing WGCNA in high-dimensional transcriptomic data. First, the TestSoftPowers function was applied to determine an appropriate soft

power threshold ($R^2 = 0.85$). After determining the soft power threshold, the ConstructNetwork function was used to create a weighted co-expression network. Subsequently, the Spearman correlation method was adopted to generate a topological overlap matrix. Then, the PlotDendrogram function was used to create a hierarchical clustering tree, thereby obtaining co-expression gene modules. The ModuleEigengenes function was employed to determine the potential feature genes associated with each gene module. The ModuleConnectivity function was used to compute the k-module membership (kME) values across the complete single-cell dataset. It filtered the genes to identify the top 200 kME values as candidate feature genes for each module. Intergroup Wilcoxon tests were conducted using FindDMEs to compare the expression level of signature genes in each module between the SSc-ILD group and normal samples ($P < 0.05$). The modules exhibiting the most significant changes in signature gene expression levels in the SSc-ILD state were identified, specifically the upregulated and downregulated modules with the largest absolute values of $\log_2$ fold change (FC). Ultimately, the feature genes with the top 200 kME values in these modules were selected as the module genes associated with the key cells.

## 2.4. Cell communication and pseudotime analysis

In analyzing the communication between key cells and other cell types, the samples from the GSE122960 dataset were used. Based on the cell-type annotation completed in the previous steps, cell communication analysis was performed using the CellChat package (v 1.6.1) [20]. Ligand–receptor interactions among key cells were visualized with bubble plots, and communication probabilities were assessed using a permutation test with 1000 random shuffles (nboot = 1000). In addition, the ligand–receptor interactions among key cells were visualized, and a bubble plot of the ligand–receptor interactions was drawn. Meanwhile, they were clustered in a descending dimension using the method described in Section 2.2 to understand the differentiation trajectory of key cells. The proportion of different subgroups of key cells in SSc-ILD samples and normal samples were visualized. Then, the Monocle package (v 2.28.0) [21] was used for pseudotime analysis, and the cell differentiation trajectory diagram was drawn.

## 2.5. Differential analysis and functional enrichment of macrophage subclusters

To further explore the functional heterogeneity of macrophages in SSc-ILD, differential expression analysis was performed between specific macrophage subclusters and the remaining macrophage population. Macrophage subclusters were identified through unsupervised clustering, as described in Section 2.2. Differential expression analysis was conducted using the Seurat package (v 5.1.0) with the following thresholds: $|\log_2 FC| > 0.25$ and adjusted $P$ value $< 0.05$. Then, the upregulated genes in each subcluster were subjected to Gene Ontology (GO) and Kyoto Encyclopedia of Genes and Genomes (KEGG) pathway enrichment analyses using the clusterProfiler package (v 4.8.3). In addition, to deeply explore the specific functional characteristics of macrophage subsets, the known functional marker gene sets were systematically integrated for targeted analysis. The gene sets used include M1/proinflammatory genes (IL1B, TNF, NOS2, and CD86), M2/profibrotic genes (ARG1, CD163, MRC1, TGFB1, SPP1, MMP9, and COL1A1), settled/homeostasis-related genes (FABP4, TIMD4, and LYVE1), and monocyte origin/recruitment-related genes (CCR2, S100A8, and S100A9). The functional phenotypic characteristics of the identified macrophage subsets were precisely characterized by visualizing the expression levels of the abovementioned marker genes.

## 2.6. Differential expression analysis

To identify DEGs between SSc-ILD and normal groups, DEGs were carried out by employing the limma package (v 3.56.2) [22] based on GSE81292 (training set) for both sample groups ($|\log_2 FC| > 0.5$, $P$adj $< 0.05$). Moreover, the volcano plot and heatmap of DEGs were produced via the ggplot2 package (v 3.5.1) [23] and the ComplexHeatmap package (v 2.16.0) [24], respectively.

## 2.7. Identification of candidate genes and functional enrichment

The ggvenn package (v 0.1.10, https://CRAN.R-project.org/package=ggvenn) was used to determine the overlap of DEGs and module genes and identify the candidate genes associated with SSc-ILD and key cells. Afterward, GO and KEGG analyses were conducted to determine the potential biological functions and mechanisms associated with the candidate genes, employing the clusterProfiler package (v 4.8.3, $P<0.05$) [25,26]. To understand the diseases related to candidate genes, Disease Ontology (DO) enrichment analysis was performed using the DOSE package (v 2.0.0) [27] with a screening condition at $P<0.05$.

## 2.8. Protein–protein interaction analysis and expression validation

The candidate genes were uploaded into the Search Tool for the Retrieval of Interacting Genes/Proteins database (https://string-db.org/; usage time: 20250226) with a confidence score of ≥0.4. After removing discrete genes, a PPI network was constructed and visualized using Cytoscape software (v 3.8.0) [28] for the candidate genes. Moreover, to identify the most significant genes among the candidate genes, the PPI networks of these candidates were analyzed using the molecular complex detection (MCODE) and maximal clique centrality (MCC) algorithms in Cytoscape software (v 3.8.0). Then, the top 10 genes identified by each algorithm were intersected to define the candidate key genes for this study.

Subsequently, an investigation was conducted to compare the expression level of the candidate key genes in the SSc-ILD and normal groups, using the GSE81292 and GSE76808 datasets. The Wilcoxon test was employed to analyze the datasets ($P<0.05$). Genes exhibiting consistent expression trends and significant intergroup differences in both datasets were identified as key genes.

## 2.9. Chromosome localization, correlation, subcellular localization, and GeneMANIA network of key genes

In this study, the RCircos package (v 1.2.2) [29] was used to determine the chromosome localization of key genes. To further analyze the functional similarity among key genes, the GO semantic similarity of these genes was calculated using the GOSemSim package (v 2.26.1) [29]. A final score of >0.5 indicates a high level of functional similarity. The subcellular localization of key genes was analyzed to obtain insights into their functions. Sequence files of key genes in FASTA format were obtained from the National Center for Biotechnology Information database (https://www.ncbi.nlm.nih.gov/gene/). Subsequently, the subcellular localization of these genes was predicted using the mRNALocater database (http://bio-bigdata.cn/mRNALocater/). Then, the results were visualized and presented using the ggplot2 package (v 3.5.1). The Gene Multiple Association Network Integration Algorithm (GeneMANIA) website (http://www.genemania.org/) was used to construct a gene co-expression network to further predict the interactions between key genes and their involved biological functions.

## 2.10. Gene set enrichment analysis (GSEA)

Initially, the Spearman correlation coefficients between the key genes and every single gene across the entire range of samples from the training set were calculated using the psych package (v 2.4.6.26) [30]. Then, the sorted genes were arranged in descending order according to their respective Spearman correlation coefficients, with this ranking being used as the gene set that was tested in further analyses. Meanwhile, the reference gene set (c2.cp.kegg.v7.4. symbols. gmt) was retrieved from the Molecular Signatures Database. Subsequently, GSEA was performed using the clusterProfiler package (v 4.8.3), with a threshold set at |normalized enrichment score |>1, adj.$P<0.05$, and false discovery rate <0.25.

## 2.11. Immune infiltration analysis

The relative infiltration levels of 22 types of immune cells [31] were quantified using cell-type identification by estimating relative subsets of the RNA transcript (CIBERSORT) algorithm in the training set. Then, the Wilcoxon test was conducted to

ascertain the immune cells exhibiting notable disparities between the two groups ($P<0.05$). The outcomes were presented in a box plot using the ggplot2 package (v 3.5.1). Subsequently, Spearman's correlation analysis was performed to investigate the relationship among the varying immune cell populations, using the psych package (v 2.4.6.26), with a threshold set at |correlation (cor)|$>0.3$ and $P<0.05$. Finally, Spearman's correlation analysis was conducted to examine the association between the key genes and differential immune cells using the psych package (v 2.4.6.26; |cor|$>0.3$, $P<0.05$).

## 2.12. Regulatory network construction

To comprehensively understand the role of upstream regulatory factors in the regulation of key genes, the Cistrome database (http://cistrome.org/) was used to predict transcription factors (TFs) associated with these genes. The TargetScan database (http://www.targetscan.org/) was employed to identify microRNAs (miRNAs) related to the key genes. Subsequently, Cytoscape software was used to visualize the prediction of TFs, mRNAs, and miRNAs.

## 2.13. Potential drug prediction and molecular docking

To identify the drugs that might interact with key genes, potential drug candidates targeting these genes were analyzed using the drug–gene interaction database (DGIdb, https://dgidb.org/). Cytoscape software (v 3.8.0) was used to visualize the gene–drug interaction network. To understand the mechanism by which key genes bind to drugs, molecular docking was performed for each gene with the highest binding score based on the drug obtained in the previous step. The structure files of the drugs were sourced from the PubChem database (https://pubchem.ncbi.nlm.nih.gov/), and the protein structures of the key genes were extracted from the Protein Data Bank (https://www.rcsb.org/). Afterward, molecular docking was conducted using the CB-Dock2 online website (https://cadd.labshare.cn/cb-dock2/php/index.php). The binding activity was considered satisfactory if the affinity was $<-5.0$ kcal/mol. Molecular docking was also performed to investigate the binding ability between cyclophosphamide and key genes.

## 2.14. Macrophage fibrosis–based subgrouping and key gene expression analysis

AUCell scoring was performed using fibrosis characteristic genes (SPP1, GPNMB, MERTK, CD9, FABP5, TREM2, CD63, PSAP, and CAPG) to explore macrophage classification by fibrosis degree. Macrophages were divided into high- and low-fibrosis subgroups based on median AUCell scores. UMAP visualization was used to map macrophage subclusters to fibrosis subgroups and analyze the expression of key genes in the two subgroups.

## 2.15. Macrophage functional scoring and correlation analysis

Based on the established set of macrophage functional characteristic genes (such as M1, M2, phagocytosis, antigen presentation, and other related gene sets), the ssGSEA method of the GSVA package was adopted to calculate the functional enrichment score of each sample in the training set (GSE81292), thereby quantifying the functional status of its macrophage population. Subsequently, Spearman rank correlation analysis was used to evaluate the correlation between the expression levels of key genes and the scores of various functions.

## 2.16. Statistical analysis

Bioinformatic analyses were conducted using the R language (v 4.3.3). A Wilcoxon test was used to assess the differences between the two groups. A $P$ value below 0.05 indicated statistical significance.

## 2.17. Ethics committee approval

The study protocol was approved by the Zhangzhou Affiliated Hospital of Fujian Medical University Ethics Committee (date: June 30, 2025, no. 2025LWB276). The study was conducted following the principles of the Declaration of Helsinki.

## 3. Results

### 3.1. Identification of macrophages as key cells

To identify the differences in cell subsets and functional states in the lung tissue of SSc-ILD at the single-cell level, clarify the core cell types involved in disease progression, and lay the foundation for subsequent mechanism research focusing on macrophages, this study conducted quality control on the original data of the GSE122960 dataset to ensure the reliability of the analysis. First, quality control was performed on the single-cell dataset GSE122960, retaining 49,160 cells and 22,377 genes (S1 Fig). The top 2000 HVGs were screened out, among which the genes with the highest variability included IGKC, SCGB1A1, and IGHM (Fig 1a). After computing 20 Principal components (PC), the significance decreased, and the curve in the PC scree plot flattened at PC = 20. Therefore, the top 20 PCs were selected for further analysis (Fig 1b, c). Second, cells were divided into 27 cell clusters using the UMAP clustering method (Fig 1d). Based on the expression of marker genes (Fig 1e), the 27 cell clusters were annotated into 12 cell types: macrophages, type I alveolar epithelial (AT1) cells, type II alveolar epithelial (AT2) cells, monocytes, endothelial cells, ciliated cells, club cells, dendritic cells, T cells, B cells, fibroblasts, and mast cells (Fig 1f).

### 3.2. Cell communication of key cells and pseudotime analysis

In this study, cell communication analysis and pseudo-temporal trajectory reconstruction were conducted to reveal the intercellular signal communication patterns and differentiation and evolution paths of macrophages in diseases, as well as explore the interaction and state dynamics of macrophages with other cells in the SSc-ILD microenvironment. Cell communication analysis revealed the connection between macrophages and other annotated cell types. The analysis showed that in the normal group, the number of macrophages was also the highest, and the communication frequency between macrophages and AT1 cells was the strongest (Fig 2a, b), whereas in the SSc-ILD group, the communication frequency between macrophages and AT2 cells was the strongest (Fig 2c, d). Whether in the SSc-ILD group or the normal group, the communication probability of the key cells from macrophages to AT2 cells was the highest, and the corresponding ligand–receptor interaction was LGALS9–CD44 (Fig 2e, f). After dimensionality reduction and clustering, macrophages were divided into 14 subgroups (Fig 2g). In subgroup 5, macrophages had the highest proportion of cells in the SSc-ILD group; in subgroup 3, macrophages had the highest proportion of cells in the normal group (Fig 2h). To confirm the developmental stages of the macrophages, pseudotime series analysis was performed. The results showed that macrophages differentiated over time from left to right. The 14 subgroups could be roughly divided into five differentiation states. Among them, states 1–3 are the early differentiation stages, and state 4 is the relatively late differentiation stage. Subpopulations 8 and 3 were identified as the earliest cell subpopulations differentiated and formed (Fig 2i).

### 3.3. Macrophage fibrosis subgroups and key gene expression characteristics

The AUCell score based on fibrotic characteristic genes successfully classified macrophages into highly fibrotic and less fibrotic macrophages (S2a Fig). Coverage analysis of macrophage subclusters and fibrotic subgroups revealed that different subclusters had different proportions of highly fibrotic macrophages. In particular, the proportion of highly fibrotic cells in macrophage subclusters 0, 5, and 7 was significantly higher than that in other subclusters (S2b Fig). UMAP visualization shows that the high expression cells of the three key genes (ARG2, ELF3, and NKX2−1) in highly fibrotic and less fibrotic macrophages are dispersed in the highly fibrotic and less fibrotic subgroups. In addition, no evident spatial aggregation was observed in the highly fibrotic subgroup (S2c–e Fig).

### 3.4. Identification of 400 macrophage module genes

In screening for the gene set that is closely related to the functional status of macrophages, the hdWGCNA method was adopted in this study to construct a gene co-expression network. In addition, the modules with the most significant

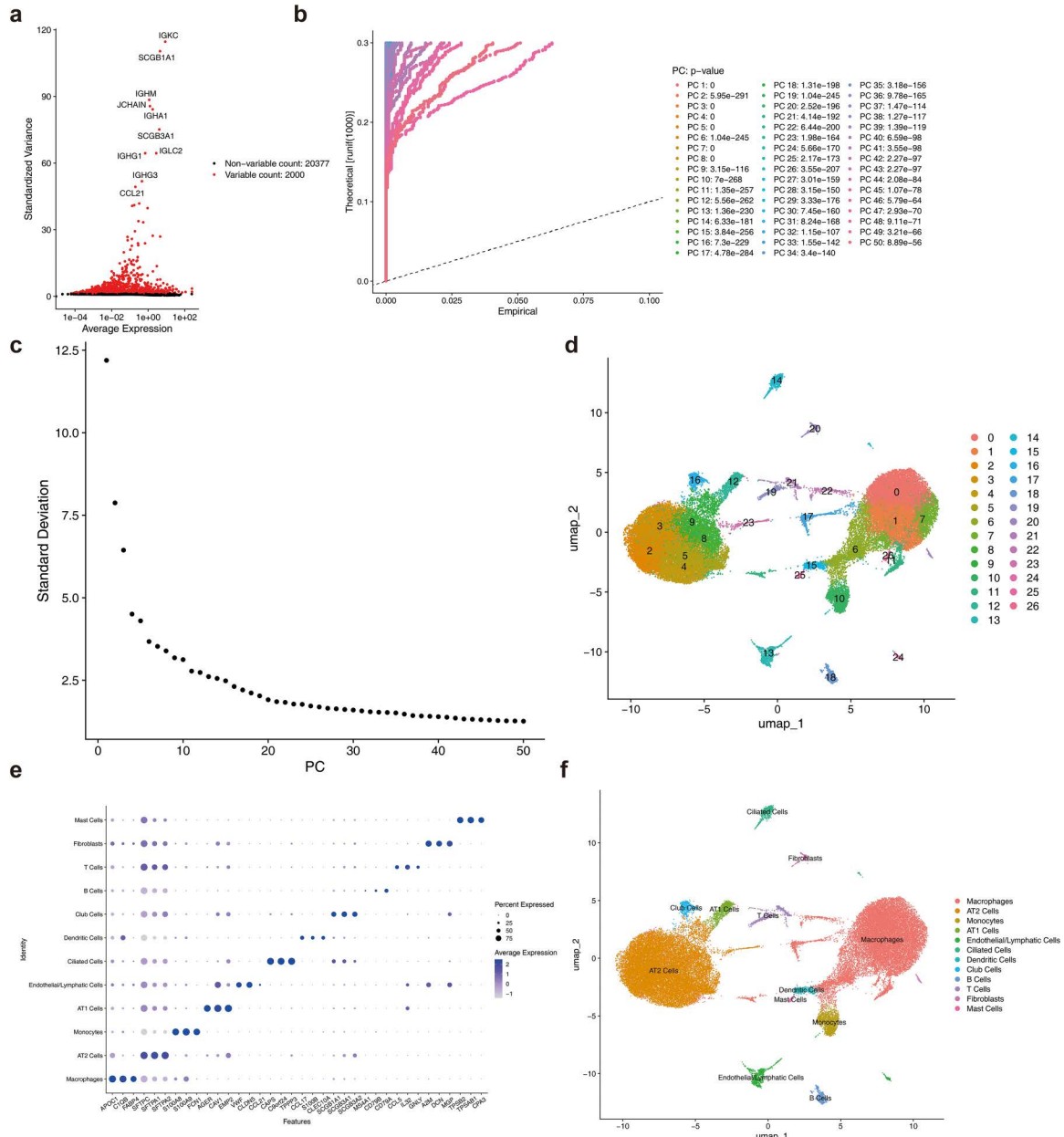

**Fig 1. Identification and analysis of key cells.** All analyses were performed on the basis of the GSE122960 dataset, and all samples in this dataset were included simultaneously. Before the analysis, all samples underwent unified quality control, standardization, and integration to ensure the comparability of the results. (a) Screening for highly variable genes. The red dots represent the top 2000 highly variable genes, while the black dots indicate the genes with smaller coefficients of variation. The top 10 highly variable genes are labeled in the figure. (b) Principal component (PC) plot based on the Jackstraw function permutation test algorithm. (c) Scree plot generated by the ElbowPlot function. (d) UMAP diagram for cell cluster classification. (e) Bubble plot of the expression of marker genes in different cells. (f) Cell annotation diagrams for different cell types. Twelve cell types were annotated for different cell clusters, with different colors representing different cell types.

expression changes under disease conditions were identified, and their core genes were extracted as the basis for subsequent association analysis. A co-expression network was constructed via hdWGCNA, with the soft-thresholding power set at 3 ($R^2 = 0.85$; S3a Fig), and eight gene modules were identified (S3b Fig). Moreover, 25 module candidate signature

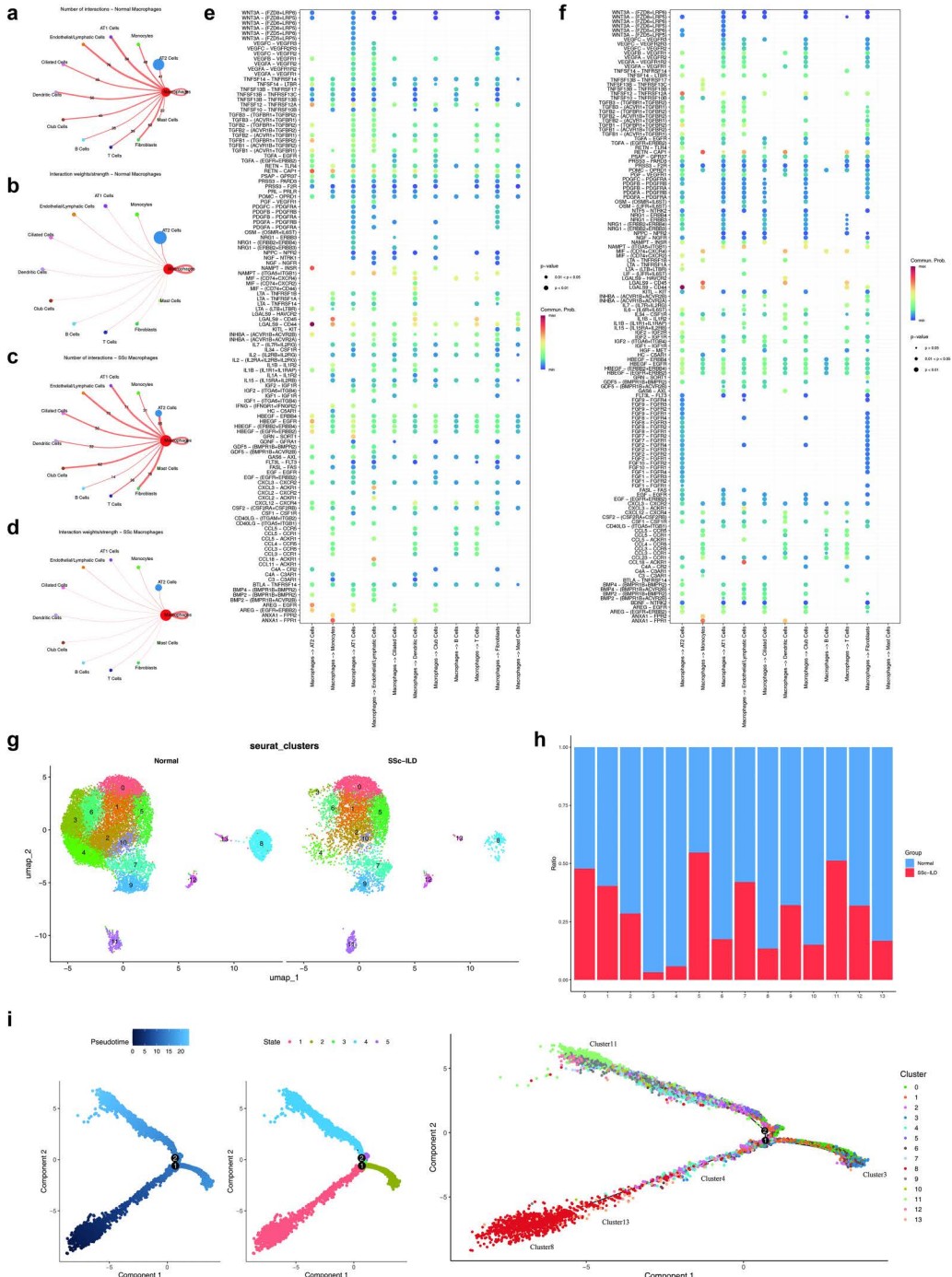

**Fig 2. Cell communication of key cell and pseudotime analysis.** (a) The number of interactions in the communication network of the control group. (b) The number of interactions in the communication network of the SSc disease group. (c) The intensity of communication interaction in the communication network of the control group. (d) Interaction intensity of communication networks in the SSc disease group. The thickness of the connection lines represents the intensity of communication. (e–f) Ligand–receptor interaction point diagram. From left to right are the ligand–receptor interaction point maps of the control and SSc disease groups, respectively. The colors of the dots from red to blue represent the intensity of communication from strong to weak, and the size of the dots represents the $P$ value. The more significant the $P$ value, the larger the dots. (g) Distribution of different subgroups of macrophages; a total of 14 subgroups were obtained from clusters 0–13. (h) Differences in the proportion of cells are found between different subgroups of macrophages in the disease and control groups, with red indicating the proportion of the SSc disease group and blue indicating the proportion of the control group. (i) The differentiation trajectory of macrophages, from left to right, consists of the cell differentiation time, cell state, and cell-type trajectories. The different colors represent the different states of cell differentiation.

genes were distributed in the eight modules within macrophages (Fig 3a). A relatively high $\log_2$ FC absolute value was observed between the brown ($\log_2$ FC = 0.40, $P < 0.5$) and blue ($\log_2$ FC = −0.54, $P < 0.5$) modules (Fig 3b). Therefore, these two modules were regarded as key modules, and the 400 genes in these modules (top 200 genes with kME values of 555 genes in the brown module and top 200 genes with kME values of 990 genes in the blue module) were used as macrophage module genes for subsequent analysis.

### 3.5. Association of 50 candidate genes with macrophages and SSc-ILD

In integrating transcriptome and single-cell module information and identifying key genes that are simultaneously associated with disease phenotypes and macrophage functions, the intersection of DEGs and macrophage module genes was taken to obtain a candidate gene set, and functional and disease enrichment analyses were conducted to explain its potential biological significance. A total of 1515 DEGs were identified via differential expression analysis, among which 870 were upregulated and 645 were downregulated (Fig 4a, b). The intersection of DEGs and macrophage module genes was obtained, yielding 50 candidate genes (Fig 4c). Enrichment analysis of the candidate genes revealed associations with 585 GO terms. These terms encompassed a wide range of functions such as the regulation of peptidase activity, ficolin-1-rich granule lumen, and activin binding (Fig 4d, S1 Table). In addition, 24 KEGG pathways were identified, including pyruvate metabolism, apoptosis, and viral carcinogenesis (Fig 4d, S2 Table). Pyruvate metabolism plays a critical role in the metabolic reprogramming of immune cells, particularly macrophages [32,33]. Its activation is generally associated with the polarization of macrophages toward a proinflammatory phenotype and may play a role in the inflammatory response and fibrotic process in SSc-ILD by influencing energy metabolism and the synthesis of inflammatory mediators. DO enrichment analysis revealed that the candidate genes were significantly correlated with cerebral ischemia, gastric cancer, cerebrovascular disease, germ cell carcinoma, embryoma, and other diseases (Fig 4e).

### 3.6. Identification of ARG2, ELF3, and NKX2−1 as key genes

To screen the key driver genes of SSc-ILD, PPI network analysis and hub gene identification were conducted on the candidate genes, combined with expression validation in independent datasets, and the core genes were finally determined. The PPI network showed that the candidate genes had 39 protein interaction relationships, such as CDH1-AGR2 and SCD1-ICAM1 (Fig 5a). PPI analysis revealed that the candidate genes had multiple networks and synergistic interactions. Subsequently, of the 50 significant genes, 10 were identified using the MCODE and MCC algorithms (Fig 5b). The significant genes identified by the MCODE and MCC algorithms completely overlapped, and they were designated as candidate hub genes: ELF3, SDC1, ICAM1, RAB25, S100A14, KRT19, CDH1, ARG2, CD63, and NKX2−1. Based on the GSE81292 and GSE76808 datasets, three of the 10 candidate hub genes exhibited significant differences and consistent trends across both datasets. In particular, the expression levels of ARG2 and NKX2−1 were significantly upregulated in the SSc-ILD group, whereas that of ELF3 was significantly downregulated in the SSc-ILD group ($P < 0.05$; Fig 5c, d). Consequently, three key genes (ARG2, ELF3, and NKX2−1) were selected for subsequent analysis.

### 3.7. Key genes located in the cytoplasm and endoplasmic reticulum were significantly enriched in multiple signaling pathways

Chromosomal localization, functional similarity analysis, subcellular localization prediction, and GSEA enrichment analysis were conducted to reveal the potential molecular mechanism involved in SSc-ILD from multiple dimensions. Chromosomal localization shows that ELF3 is located on chromosome 1, and ARG2 and NKX2−1 are on chromosome 14 (S4a Fig). The association of function between the biomarkers was explored, and the results indicated that the functional similarity between NKX2−1 and ELF3 was higher ($P > 0.5$; S4b Fig). In addition, NKX2−1 was primarily located at the endoplasmic reticulum, whereas ELF3 and ARG2 were predominantly found in the cytoplasm (Fig 6a). In the GeneMANIA network, the

 

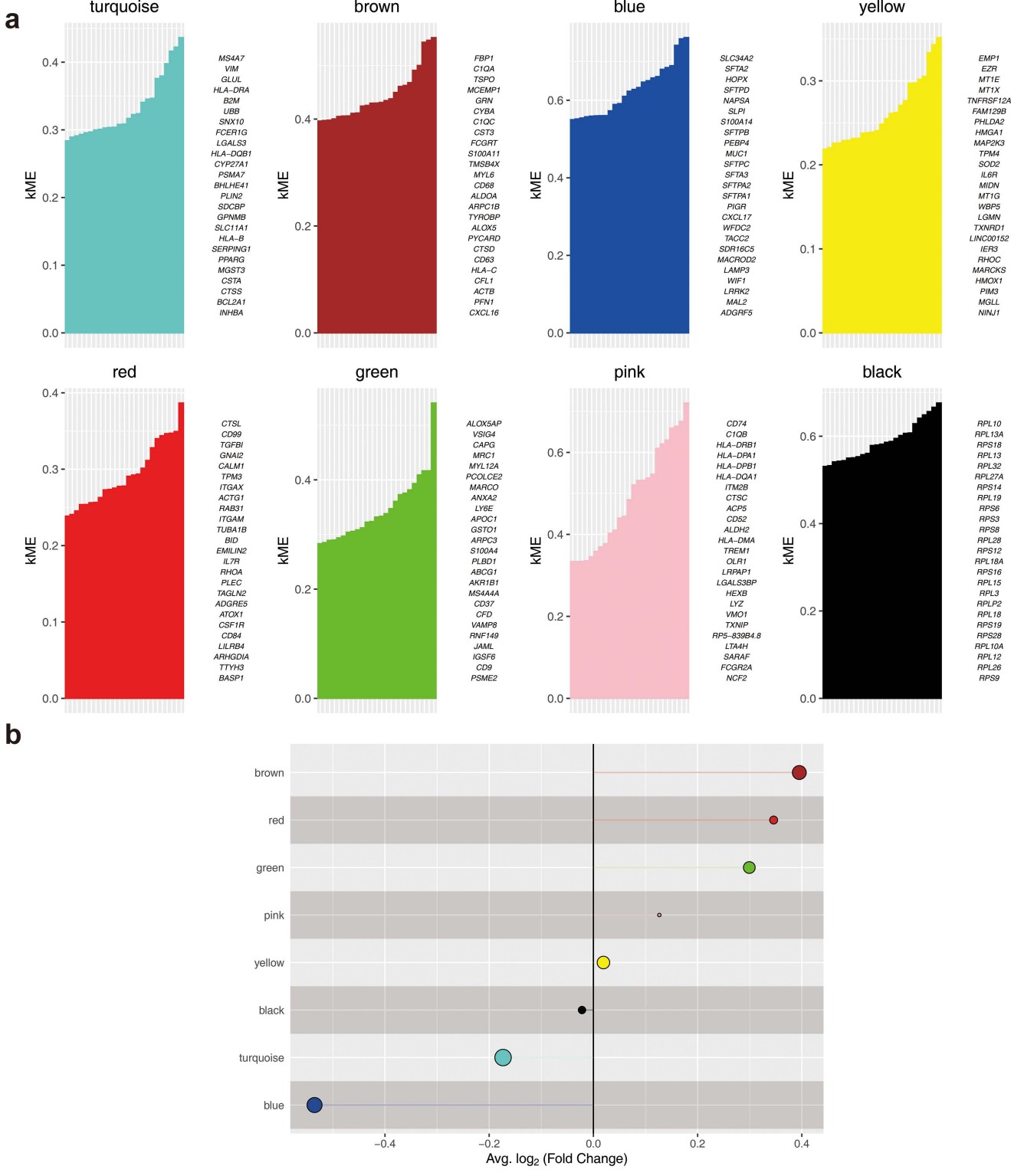

**Fig 3. hdWGCNA identifies the module genes of macrophages.** (a) Main candidate feature genes for different modules. The top 25 genes with the highest pairwise correlation between genes and module feature genes in each module. (b) The differences in candidate feature gene expression levels between the SSc-ILD group and the control group. The size of the circle represents the number of genes in the module, and the horizontal axis represents the different multiple log$_2$FC values.

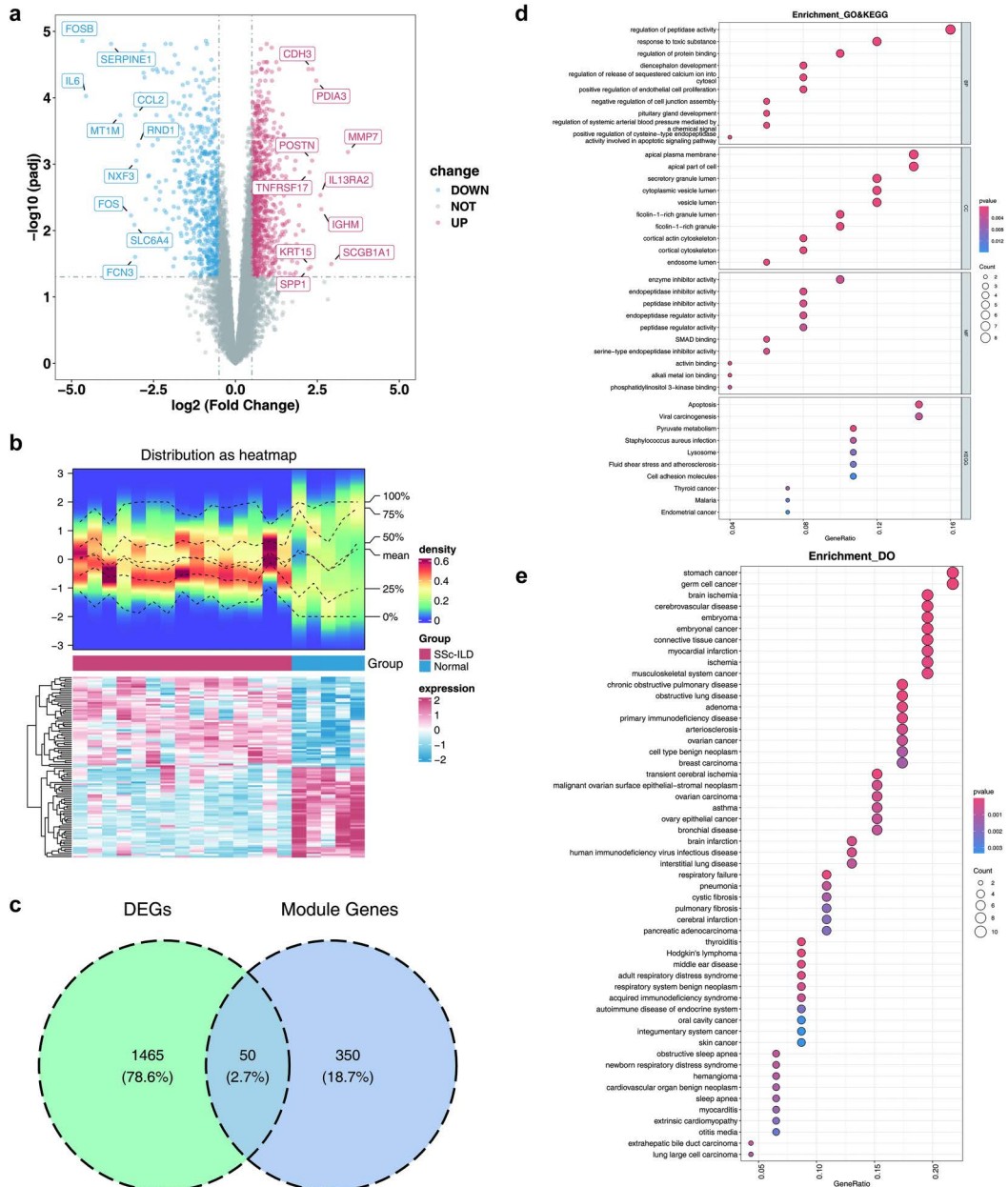

**Fig 4. Identification of differentially expressed and candidate genes.** (a) The volcano map of DEGs. The horizontal axis represents the $\log_2$Fold-Change, and the vertical axis represents −log10 ($P$adj), with each point representing a gene. The horizontal reference line represents $P$adj=0.05, and the vertical reference line represents $\log_2$FoldChange=±0.5. Divided by reference lines, the red dots represent the upregulated differentially expressed genes, while the blue dots represent the downregulated differentially expressed genes. The genes labeled in the figure are the top 10 upregulated genes and top 10 downregulated genes sorted by $\log_2$FC. (b) Heat map of DEGs. The top is the density distribution heatmap of differentially expressed genes, and the bottom is the heatmap of differentially expressed gene expression levels, showing the gene expression and clustering of the top 50 upregulated genes and top 50 downregulated genes sorted by $\log_2$FC. The color bars represent the relative changes in gene expression levels, from blue (low expression) to red (high expression). (c) Identification of candidate genes. The green circle represents DEGs; the blue circle represents the key module genes (Mes) obtained from WGCNA, and the overlapping area in the middle represents the genes that are present in both gene sets simultaneously. (d) GO and KEGG enrichment analyses. The size of the dots represents the number of enriched genes, with larger dots indicating more enriched genes. The color of the dot represents $P$ value, with redder dots indicating more significant enrichment result. (e) DO enrichment results. The size of the dots represents the number of enriched genes, with larger dots indicating more enriched genes.

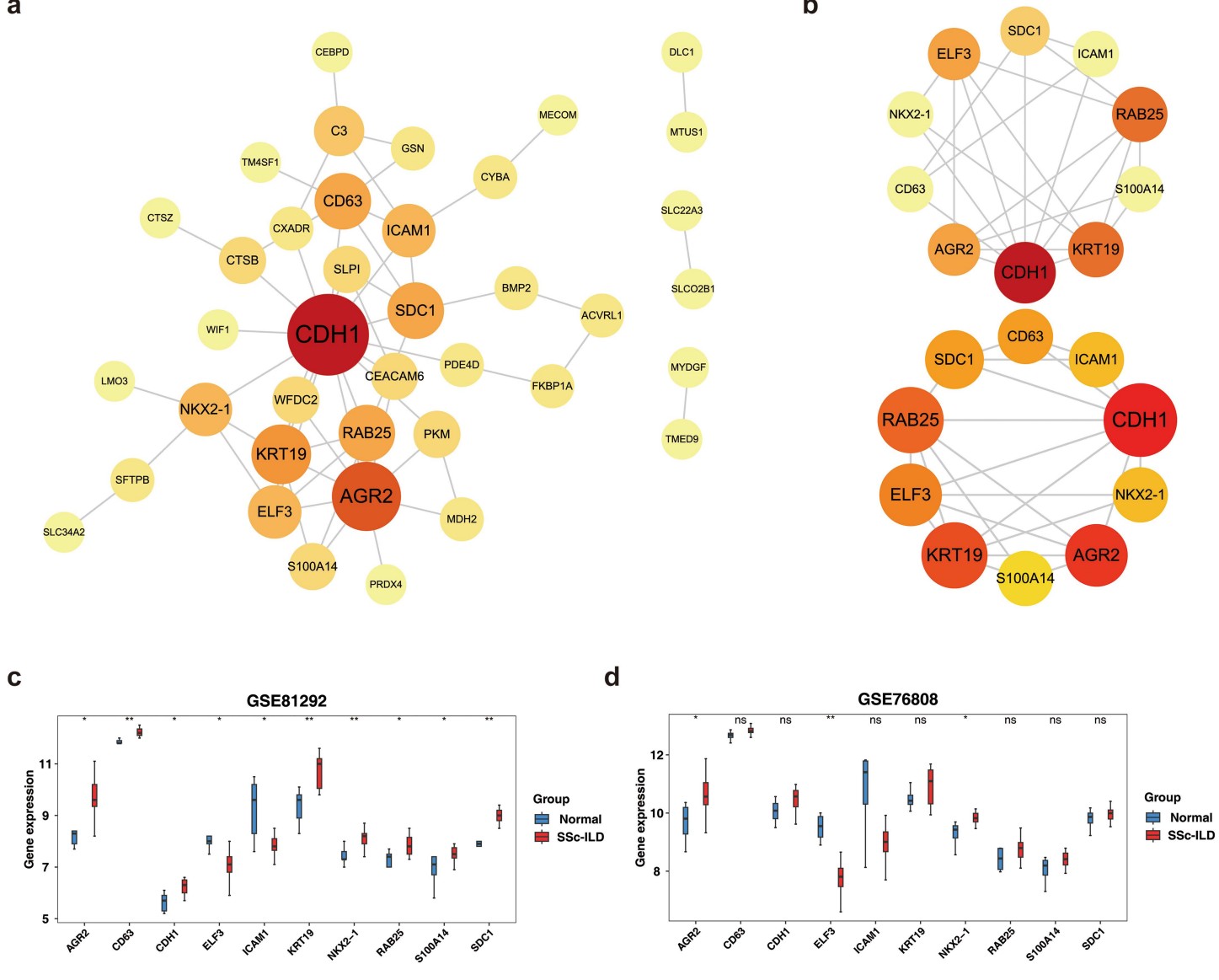

**Fig 5. Identification of key genes.** (a) Protein interaction network. The nodes represent the candidate genes, and the color of the nodes from red to yellow represents the degree of the node, which is the number of interactions between each gene and other genes. The redder the color, the more interactions the gene has and the higher the metric value. The size of the node also represents its degree. The larger the size, the more frequent the interaction between the gene and the larger the degree. (b) The important modules selected by MCODE and the hub genes obtained from MCC algorithm. (c) The expression of hub genes in the training set. The box plot indicates the differential expression of key genes in the training set, with red representing the SSc group and blue representing the normal group. $P < 0.05$ is considered statistically significant. (d) The expression of hub genes in the validation set.

key genes interacted with 20 genes, such as ARG1, HOXB3, and FOXF1, and they played a role in the arginine metabolic process, embryonic organ morphogenesis, and glutamine family amino acid metabolic process (Fig 6b).

Subsequently, GSEA analysis revealed several related signaling pathways and potential biological mechanisms associated with the key genes. In particular, ARG2, ELF3, and NKX2−1 were each enriched in 10 pathways. All key genes were co-enriched in the lysosome (Fig 6c–e). ARG2 and ELF3 were also co-enriched in the pathways of allograft rejection and graft-versus-host disease.

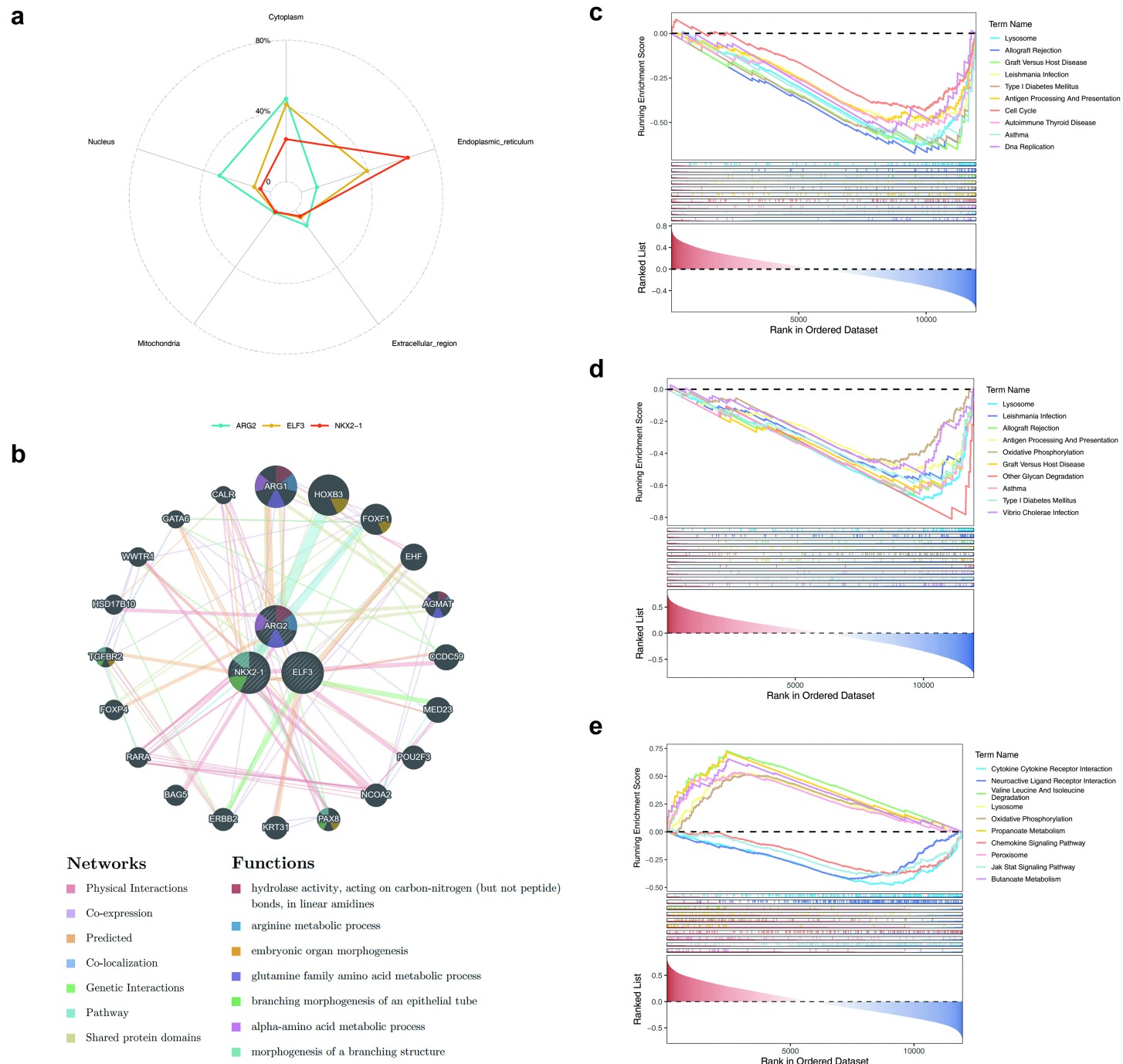

**Fig 6. Chromosomal localization, functional similarity analysis, and subcellular localization of key genes.** (a) Subcellular localization. The radar map localization points are the subcellular localization distribution probabilities predicted for each key gene, with blue representing ARG2, yellow representing ELF3, and orange representing NKX2−1. (b) GeneMANIA analysis of key genes. (c) GSEA enrichment analysis results of the key gene ARG2. The result chart is divided into three parts from top to bottom: the first part is the enrichment score (ES), with the horizontal axis representing the tested gene set sorted by correlation coefficient and the vertical axis representing the corresponding running ES. The peak of the line chart represents the ES of the enriched pathway, and the genes before the peak are the core genes located in the enriched pathway of the tested gene set. The second part is "hit," which marks the genes located under the test gene set with lines. The vertical lines are concentrated at the front or back of the gene sorting list, indicating that the gene set pathway is upregulated or downregulated. If the vertical lines are evenly distributed in the gene sorting list, then the gene set pathway has no significant changes in the two compared data. The third part is the rank value distribution map of all genes, and the Signal2Niose algorithm is used by default. (d) GSEA enrichment analysis results of the key gene ELF3. (e) GSEA enrichment analysis results of the key gene NKX2−1.

### 3.8. Correlation between key genes and immune cells

In this study, the CIBERSORT algorithm was used to quantify the infiltration level of immune cells and analyze its correlation with key genes, thereby clarifying the regulatory relationship between the key genes and the immune microenvironment and revealing the potential role of key genes in immune regulation. The CIBERSORT algorithm was also used to evaluate and quantify the infiltration abundance of 22 distinct immune cells in two designated groups: the SSc-ILD group and the normal group. S5 Fig shows a graphical representation of the infiltration abundance percentage of each immune cell across the entire sample set. As shown in the box plot, the levels of activated dendritic cells, activated mast cells, monocytes, resting natural killer (NK) cells, and follicular helper T cells were significantly reduced in the SSc-ILD group compared with the normal group. By contrast, M0 macrophages and plasma cells exhibited a high level of abundance ($P<0.05$; Fig 7a). The strongest positive correlation was found between the activated dendritic cells and resting NK cells ($r=0.67$, $P<0.01$), and the strongest negative correlation was found between the activated resting NK cells and M0 macrophages (cor = −0.60, $P<0.01$; Fig 7b). Moreover, correlation analysis revealed that the key genes were all significantly associated with seven different immune cells (Fig 7c). Among them, NKX2−1 was considerably and negatively relevant to monocytes ($r=−0.50$, $P<0.05$), whereas ELF3 and ARG2 were all significantly and positively associated with the activated dendritic cells ($r>0.60$, $P<0.01$).

### 3.9. Identification of an interaction network among key genes, TFs, and miRNAs

To reveal the upstream TFs and miRNAs that regulate the key genes, construct the TF–mRNA–miRNA regulatory network, and clarify the upstream regulatory mechanism of key genes in SSc-ILD, analysis was conducted through database prediction and network construction. The TF–mRNA–miRNA regulatory network included 211 nodes and 254 edges (Fig 8). Among them, ARG2 predicted 85 TFs and 16 miRNAs; ELF3 predicted 73 TFs and nine miRNAs; NKX2−1 predicted 64 TFs and six miRNAs. In particular, three key genes were simultaneously regulated by eight TFs, including CEBPB, RAD21, MYC, MAX, POLR2A, SMC1A, SMC, and AR. However, no common miRNAs were predicted for the three key genes.

### 3.10. Binding of key genes to SKA-111, voruciclib, and cyclophosphamide

To screen for potential therapeutic drugs targeting key genes, verify the binding activity of drugs to key genes through molecular docking, evaluate the potential therapeutic value of drugs, and provide candidate drugs and experimental basis for the targeted therapy of SSc-ILD, drug prediction and molecular docking analysis were conducted in this study. As shown in Fig 9a, potential candidate drugs targeting three key genes (ARG2, NKX2–1, and ELF3) were screened out, including 21 drugs (e.g., SKA-11 and EBIO) that targeted ARG2 and two drugs (compound 33, voruciclib) that targeted ELF3. Among them, the drugs with the highest binding scores for ARG2 and ELF3A were SKA-111 and voruciclib, respectively. The binding of SKA-111 and voruciclib to ARG2 and ELF3 was further evaluated. Based on the molecular docking results, the docking binding energies of the ARG2-SKA-111/cyclophosphamide and ELF3-voruciclib/cyclophosphamide were all less than −5 kcal/mol (Table 1). The perfect conformation indicated that SKA-111, cyclophosphamide, and voruciclib interacted with the residues of the key genes through hydrogen bonds (Fig 9b). The results of this study indicated that these pharmaceuticals might have a therapeutic effect on SSc-ILD by targeting key genes.

### 3.11. Analysis of functional characteristics of macrophage subsets 3 and 5

The transcriptional profiles of macrophage subpopulations 3 and 5 were compared with all other macrophages to elucidate the functional diversity of macrophages in SSc-ILD. In cluster 3, a total of 3121 DEGs were identified, with 1154 upregulated and 1967 downregulated genes, among which the top 10 upregulated genes were displayed (Fig 10a). Enrichment analysis of the upregulated DEGs revealed 5320 biological process (BP) terms, 956 molecular function (MF)

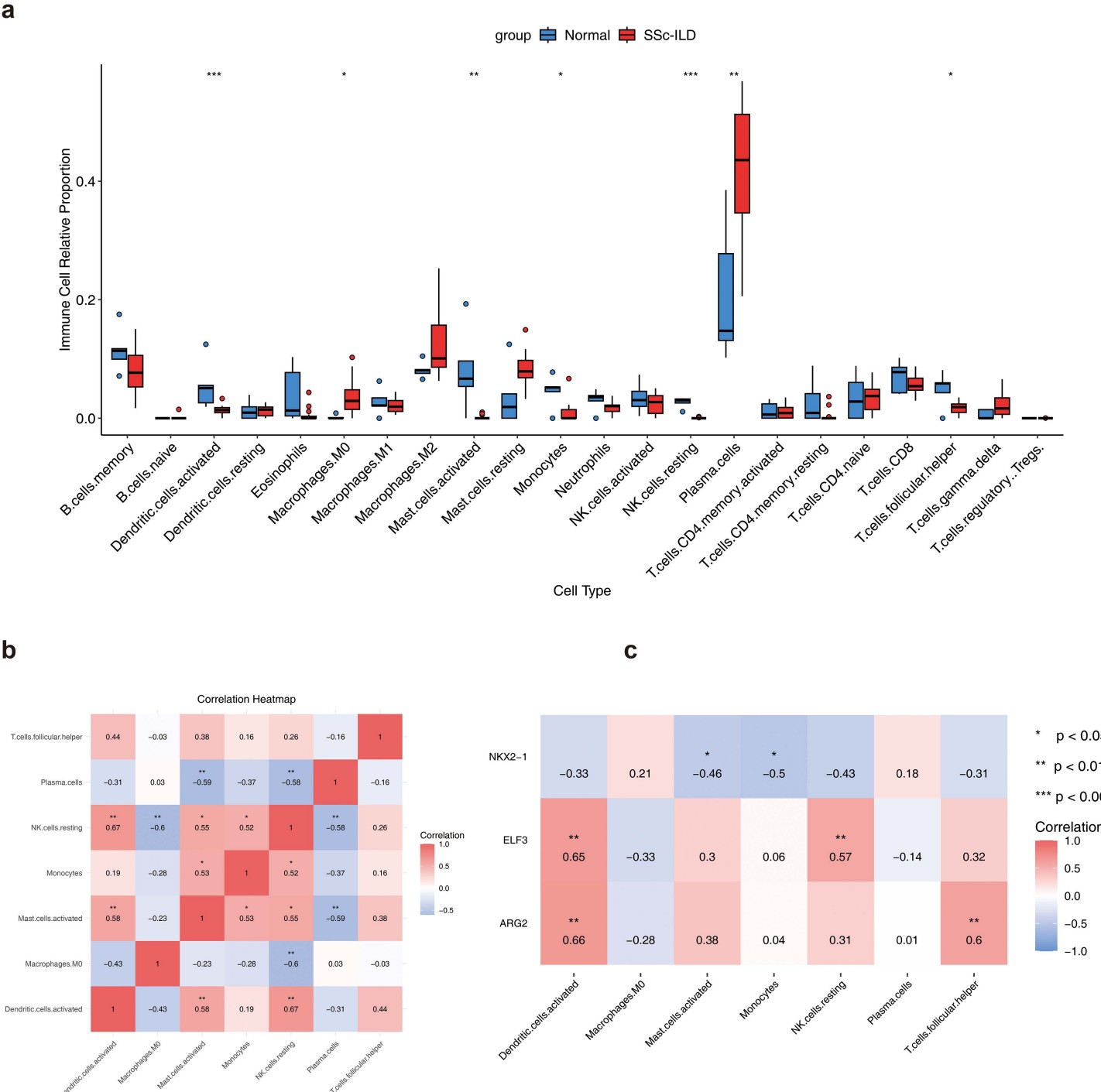

**Fig 7. Analysis of immune cell infiltration.** (a) Differential analysis of immune cells between the SSc-ILD group and the control group. The box plot indicates the differential expression of key genes in the validation set, with red representing the SSc group and blue representing the normal group. * indicates $P < 0.05$; ** indicates $P < 0.01$; *** indicates $P < 0.001$. (b) Correlation analysis between differential immune cells. The labeled value represents the intercellular correlation coefficient cor, where cor > 0 is red and cor < 0 is blue. * indicates $P < 0.05$; ** indicates $P < 0.01$. (c) Correlation analysis between differential immune cells and key genes. The asterisk indicates the significant $P$ value of gene cell correlation, where cor > 0 is red and cor < 0 is blue. * indicates $P < 0.05$; ** indicates $P < 0.01$.

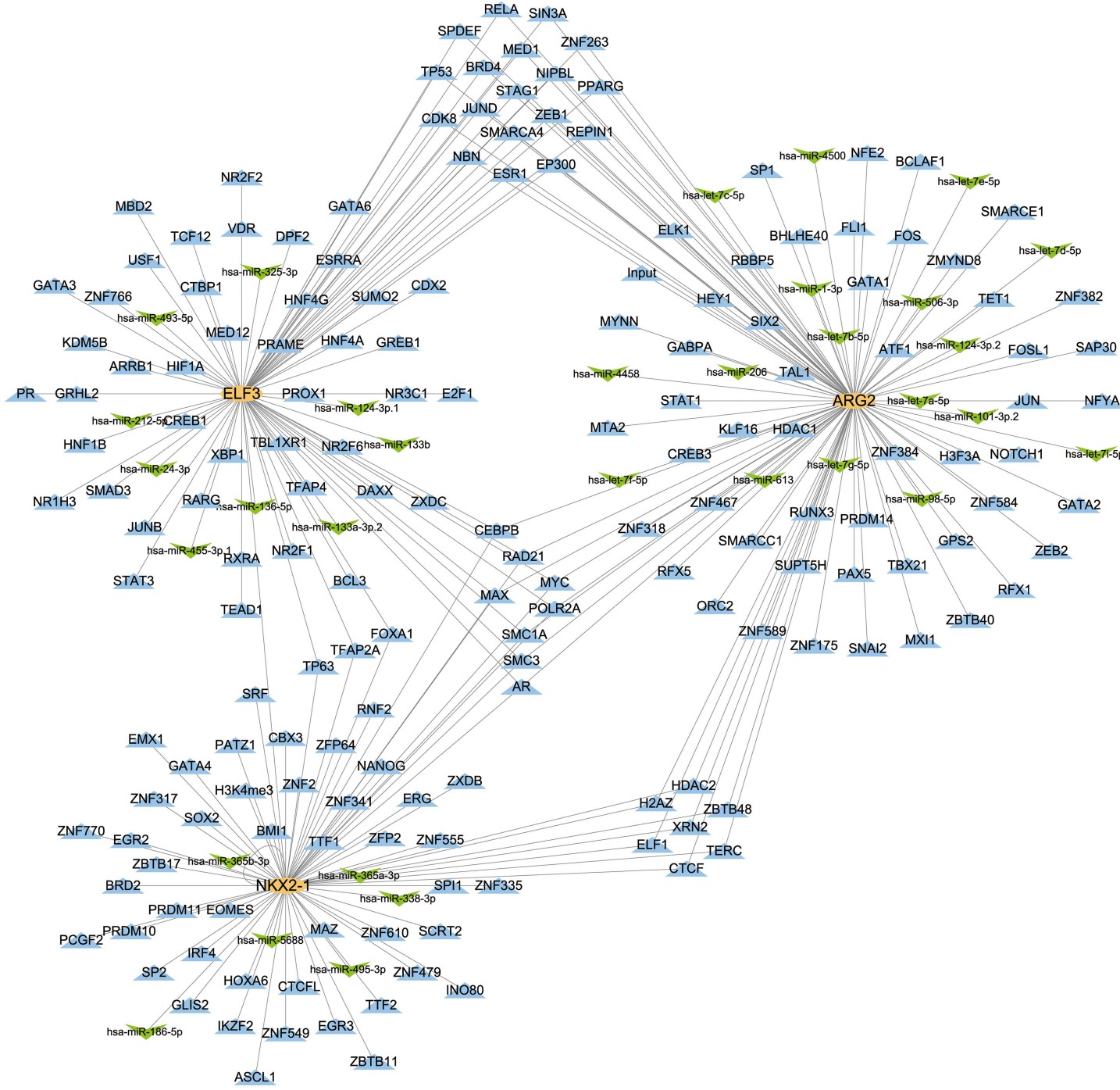

**Fig 8. Interaction network identified between key genes, TFs, and miRNAs.** TF–mRNA–miRNA regulatory network. Yellow represents the key genes; green represents the predicted TFs; and blue represents the predicted miRNAs.

a

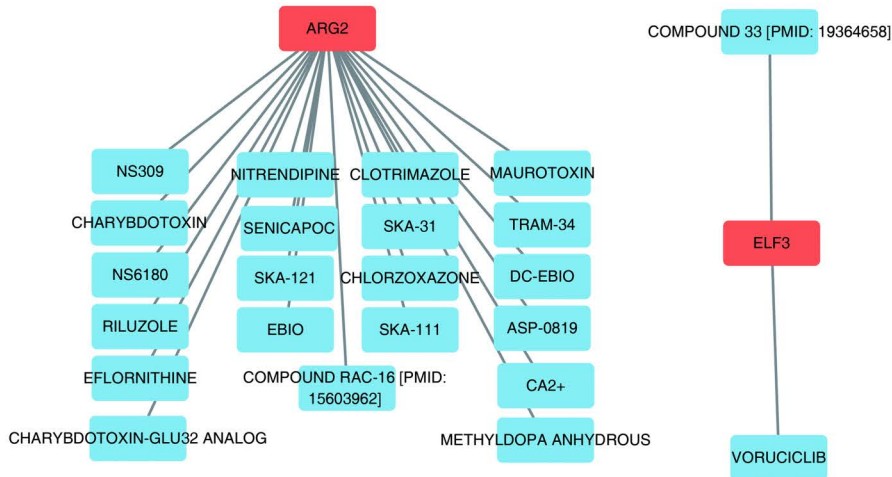

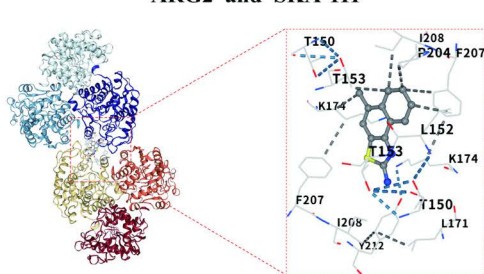

b

**ELF3 and VORUCICLIB**

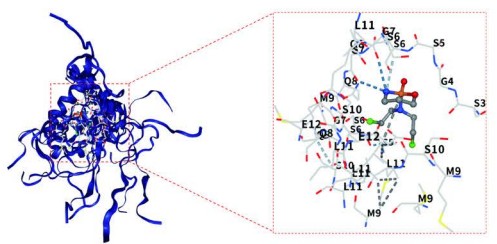

**ARG2 and SKA-111**

**ELF3 and Cyclophosphamide**

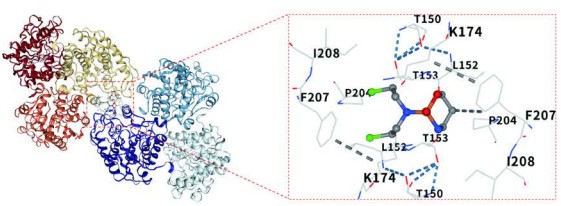

**ARG2 and Cyclophosphamide**

**NKX2-1 and Cyclophosphamide**

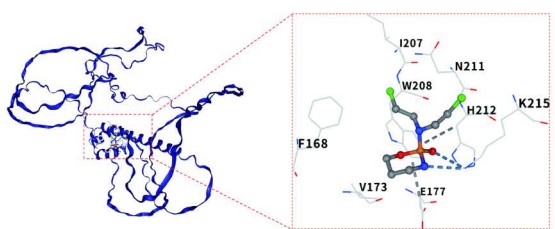

**Fig 9. Drug prediction and molecular docking.** (a) The drug prediction network diagram of ARG2 and ELF3. Red squares represent genes, and blue squares represent related drugs. (b) The molecular docking results of ARG2, ELF3, and NKX2−1 with drugs. The left part is the overall graph, and the right part is the locally magnified graph.

**Table 1. Binding energies of the molecular docking.**

| GENE | PDB ID | Drug | Binding energy (kcal/mol) |
|---|---|---|---|
| ARG2 | 1PQ3 | SKA-111 | −9.2 |
| ELF3 | 2E8P | VORUCICLIB | −8 |
| ELF3 | 2E8P | Cyclophosphamide | −8.1 |
| ARG2 | 1PQ3 | Cyclophosphamide | −6 |
| NKX2−1 | AF-P43699-F1 | Cyclophosphamide | −4.7 |

terms, 601 cellular component (CC) terms, and 330 KEGG pathways. Biological processes were significantly enriched in viral processes, RNA splicing, the regulation of response to biotic stimulus, response to virus, and mRNA processing. CC terms were significantly enriched in nuclear speckles, MHC protein complex, spliceosomal complex, MHC class II protein complex, and catalytic step 2 spliceosome. MFs were enriched in MHC class II protein complex binding, RNA polymerase II-specific DNA-binding of TFs, DNA-binding of TFs, cadherin binding, and transcription corepressor activity (Fig 10b–d). KEGG pathway analysis revealed enrichment in influenza A, type I diabetes mellitus, antigen processing and presentation, EB virus infection, and tuberculosis pathways (Fig 10e).

In cluster 5, 2535 DEGs were identified, including 1014 upregulated and 1521 downregulated genes. The top 10 upregulated genes were displayed (Fig 10f). Enrichment analysis of the upregulated DEGs yielded 5002 BP terms, 910 MF terms, 576 CC terms, and 329 KEGG pathways. Biological processes were mainly enriched in the positive regulation of cell activation, the positive regulation of leukocyte activation, the positive regulation of lymphocyte activation, leukocyte-mediated immunity, and antigen processing and presentation. CCs were primarily enriched in the cytoplasmic vesicle lumen, vesicle lumen, secretory granule lumen, fibrinogen-1-rich granules, and fibrinogen-1-rich granule lumen. MFs were enriched in monosaccharide binding, oxidoreductase activity acting on the donor CH-OH group, carbohydrate binding, MHC class II protein complex binding, and MHC protein complex binding (Fig 10g–i). KEGG pathway analysis revealed enrichment in the phagosome, lysosome, tuberculosis, antigen processing and presentation, and glycolysis/gluconeogenesis pathways (Fig 10j).

A systematic analysis of known macrophage functional markers further delineated the distinct functional phenotypes of clusters 3 and 5. The results showed that cluster 5 was characterized by a significantly high expression level of monocyte-derived markers (S100A8 and S100A9) and proinflammatory markers (CD86). Classical M2/profibrotic markers (CD163, MRC1, and TGFB1) were also enriched in this subpopulation. By contrast, cluster 3 exhibited a prominent M2/profibrotic signature, with the expression level of M2-related markers (CD163, MRC1, and TGFB1) being significantly higher than that of other macrophages (S6 Fig). These findings indicated a significant functional polarization among macrophage subpopulations within the SSc-ILD lung tissue.

### 3.12. Correlation between key genes and macrophage functional scores

Enrichment analysis of macrophage functional scores revealed a significant difference in the M1 proinflammatory score between the SSc-ILD group and the normal group (S7a Fig). Correlation analysis revealed that NKX2−1 was significantly and negatively correlated with M1 proinflammatory and migration chemotaxis scores. ARG2 exhibited a significantly negative correlation with phagocytosis and antigen presentation scores. ELF3 showed a significantly negative correlation with phagocytosis, lipid metabolism, and antigen presentation scores (S7b Fig).

### 4. Discussion

SSc-ILD is characterized by interstitial lung inflammation and advancing fibrosis. Macrophages are recognized as pivotal regulators within the immune system, and they could play a fundamental role in promoting pulmonary fibrosis [34]. In this

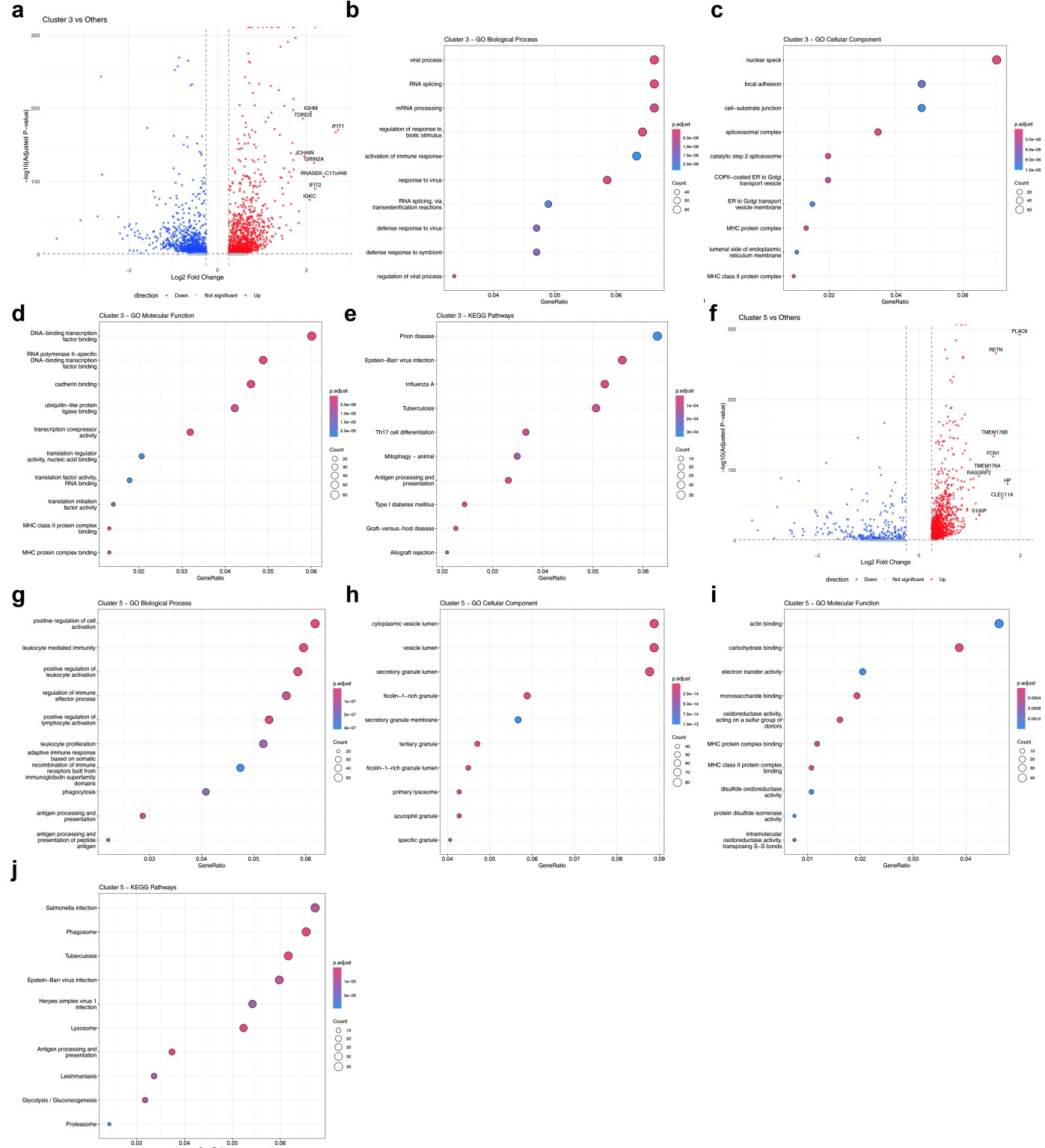

**Fig 10. Analysis of the functional characteristics of macrophage subsets.** (a) Cluster 3 difference analysis. Blue represents the genes downregulated in cluster 3, and red represents the genes upregulated in cluster 3. (b) Biochemical processes in GO enrichment analysis of cluster 3. (c) Molecular function in GO enrichment analysis of cluster 3. (d) Cellular component in GO enrichment analysis of cluster 3. (e) Enrichment results of KEGG in cluster

3. (f) Cluster 5 difference analysis. Blue represents the genes downregulated in cluster 5, and red represents the genes upregulated in cluster 5. (g) Biochemical processes in GO enrichment analysis of cluster 5. (h) Molecular function in GO enrichment analysis of cluster 5. (i) Cellular component in GO enrichment analysis of cluster 5. (j) Enrichment results of KEGG in cluster 5.

study, single-cell and batch transcriptomic data were integrated to systematically investigate macrophage-related molecular mechanisms in SSc-ILD, thereby successfully identifying three key genes: ARG2, ELF3, and NKX2−1. This study focused on the core association between macrophages and the aforementioned key genes, delving into their intrinsic link to the pathogenesis of SSc-ILD. In addition, the core value of the research was elucidated, and a more targeted theoretical support for clinical translation was provided. Furthermore, the biological pathways involving these key genes and the potential molecular regulatory networks associated with immune infiltration were dissected, thereby paving new directions for SSc-ILD treatment.

The ARG2 gene encodes arginase II, which is a key enzyme involved in arginine metabolism. ARG2 is expressed in many tissues, especially in the liver, kidney, small intestine, and macrophages. In macrophages, the expression of ARG2 can be induced by various cytokines and microbial products [35]. In immune cells, such as macrophages and lymphocytes, ARG2 expression can be induced by inflammatory signals. ARG2 plays an important role in immune response and inflammation by regulating arginine levels, thereby affecting the proliferation and differentiation of immune cells and cytokine production [36,37]. For example, in the case of infection and inflammation, the expression level of ARG2 in macrophages is upregulated, which can consume arginine in the surrounding environment and inhibit the proliferation and function of T cells, thereby regulating the intensity of the immune response [38]. In pulmonary fibrosis, the increased expression level of ARG2 correlates with enhanced fibrosis. For example, in an aged mouse model, ARG2 deficiency reduced age-related pulmonary fibrosis [39]. Furthermore, the expression level of ARG2 was upregulated in SSc-ILD, and it exhibited a positive correlation with activated dendritic cells, indicating its role in regulating the immune microenvironment and potential promotion of a profibrotic state.

The ELF3 gene encodes the E74-like ETS TF 3, which is a member of the ETS TF family. This gene modulates the expression of various target genes by binding to DNA, thereby playing a crucial role in numerous BPs, including cell proliferation, differentiation, apoptosis, migration, and immune response [40]. ELF3 is crucial for cell proliferation and differentiation. Its influence on the rate of cell proliferation is mediated by regulating the genes associated with the cell cycle [41]. Although its direct role in SSc-ILD remains unproven, our data reveal its downregulation and positive correlation with activated dendritic cells, indicating a regulatory function in immune cell activity. ELF3 may influence macrophage differentiation or cytokine production, thereby affecting inflammatory and fibrotic processes. Its enrichment in pathways such as allograft rejection further indicates its involvement in dysregulated immune responses, which are a characteristic of autoimmune diseases.

The NKX2−1 gene is responsible for encoding the NK2 family homeobox TF 1. This TF is crucial for the development and differentiation of various organs, including the lung, thyroid, and nervous system, throughout embryonic development [42]. NKX2−1 plays a key role in lung development by positively regulating the expression of alveolar surfactant protein genes and ensuring the proper development of lung tissue endings. Abnormal NKX2−1 expression is associated with lung diseases such as lung adenocarcinoma and pulmonary alveolar proteinosis [43]. The TF encoded by the NKX2−1 gene is crucial for the synthesis and metabolism of pulmonary surfactant (PS) [44]. The mutation or dysfunction of NKX2−1 can lead to PS synthesis disorders, which cause lung diseases such as respiratory distress syndrome, ILD, pulmonary fibrosis, and frequent respiratory tract infections [45]. This characteristic is similar to the pathological features of SSc-ILD. In addition, abnormal NKX2−1 gene expression may cause alveolar structural damage and fibrosis by impairing alveolar epithelial cell function, thereby exacerbating SSc-ILD [46]. Our data indicate that the expression level of NKX2−1 is upregulated in SSc-ILD, and it shows a strong negative correlation with monocytes. Therefore, NKX2−1 may contribute

to alveolar epithelial dysfunction and fibrotic response by affecting epithelial–macrophage crosstalk or surfactant metabolism.

The analysis of cellular communication in SSc-ILD revealed significantly enhanced interactions between macrophages and AT2 cells in lung tissue, with LGALS9–CD44 ligand–receptor-mediated signaling being particularly prominent. Therefore, macrophages may participate in disease progression by directly regulating AT2 cell function. In SSc-ILD, macrophages exhibit marked functional polarization alterations: On the one hand, the proportion of monocyte-derived profibrotic alveolar macrophages in bronchoalveolar lavage fluid (BALF) correlates closely with disease severity. These cells drive fibrosis by secreting key cytokines [47], and they are considered potential biomarkers and therapeutic targets [48]. Conversely, the presence of mixed-phenotype macrophages in BALF, which are characterized by imbalanced M1/M2 polarization, may exacerbate inflammation and impede repair [49]. As key cells maintaining alveolar structure and function, AT2 cells undergo dysfunction or apoptosis under injury conditions, releasing specific glycoproteins such as KL-6 into the bloodstream [50–52]. Serum KL-6 levels show significant correlations with radiographic severity and pulmonary function decline in SSc-ILD, serving as a sensitive marker reflecting alveolar epithelial injury [52,53].

Fibroblast activation, proliferation, and differentiation into myofibroblasts constitute the core pathophysiological mechanisms in SSc-ILD [54–56]. Among these, myofibroblasts, as key effector cells in ECM remodeling within SSc-ILD, play a dominant role in pulmonary fibrosis [54]. They continuously produce large amounts of ECM components such as collagen, leading to the destruction of lung tissue structure and loss of function [57–59]. Studies have confirmed a significant increase in myofibroblasts within the lung tissue of patients with SSc-ILD [54]. Single-cell analysis has revealed the heterogeneity of fibroblast populations in healthy versus SSc-ILD lung tissue, defining the transcriptional characteristics of myofibroblasts and providing crucial insights into cellular alterations during disease progression [54]. Furthermore, lung fibroblasts and skin fibroblasts exhibit distinct responses to fibrotic growth factors, indicating that fibrosis progression and fibroblast regulation may depend on tissue-specific local factors [60].

In summary, macrophages, AT2 cells, and fibroblasts collectively contribute to the inflammatory initiation, epithelial injury, and fibrotic progression of SSc-ILD through intricate intercellular interactions and signaling regulatory networks. This finding provides a crucial cellular-level theoretical foundation for further exploration of disease mechanisms and the development of targeted therapeutic strategies.

The GSEA results indicated that three key genes, namely, ARG2, ELF3, and NKX2–1, were enriched in lysosome, oxidative phosphorylation (OXPHOS), and other pathways. In SSc-ILD, the lysosomal function of immune cells such as macrophages may be impaired. After ingesting foreign antigens through phagocytosis, macrophages process and degrade them into lysosomes. Then, the antigen information is presented to T cells to initiate an immune response [61]. Lysosomal dysfunction can affect antigen processing and presentation, resulting in the abnormal activation of the immune system, thereby damaging lung tissue and promoting the development of SSc-ILD [62]. In addition, abnormalities in the electron transport chain during OXPHOS can increase the production of reactive oxygen species (ROS). In ILD, changes in the function of OXPHOS-related enzymes or mitochondrial structural damage can result in increased electron leakage during electron transfer and excessive ROS production. The excessive production of ROS can cause oxidative damage to intracellular biomolecules such as DNA, proteins, and lipids, thereby causing cell dysfunction and apoptosis [63]. Meanwhile, ROS can also act as a signaling molecule to activate a series of signaling pathways related to fibrosis, promote fibroblast proliferation and ECM synthesis, and accelerate pulmonary fibrosis. The irregular OXPHOS of ROS could trigger inflammation-associated TFs. This activation subsequently enhances the expression and secretion of inflammatory mediators, particularly IL-6 and tumor necrosis factor-α. These inflammatory factors will further recruit inflammatory cells to the lung tissue, aggravate the inflammatory response, form a vicious inflammation–oxidative stress–fibrosis cycle, and promote SSc-ILD progression [31].

In addition, KEGG enrichment analysis revealed that candidate genes were significantly enriched in the pyruvate metabolism pathway. Recent studies have progressively revealed that the metabolic state of immune cells, particularly

macrophages, is closely associated with their functional phenotypes [64–66]. Metabolism plays a central role in linking glycolysis to the tricarboxylic acid (TCA) cycle, and the dynamic regulation of its metabolic flux directly influences the functional polarization of macrophages. This correlation is particularly crucial in establishing and maintaining the proinflammatory M1-like phenotype. Studies have indicated that when stimulated by lipopolysaccharide, IFN-γ, and other factors, macrophages rapidly initiate metabolic reprogramming. This phenomenon manifests as enhanced glycolysis, suppressed mitochondrial OXPHOS, and disrupted TCA cycle activity, thereby forming an "immune metabolic switch" [67]. In SSc-ILD, the metabolic reprogramming of macrophages may exacerbate the local inflammatory microenvironment via the pyruvate metabolic pathway, thereby promoting fibroblast activation and ECM deposition and accelerating the progression of pulmonary fibrosis. This study preliminarily suggests that pyruvate metabolism may be implicated in macrophage function within SSc-ILD; however, its precise regulatory mechanism and potential as a therapeutic target require further experimental validation.

Immune infiltration analysis further reinforced the association between key genes and macrophage-centered immune regulation: ELF3 and ARG2 showed strong positive correlations with activated dendritic cells ($r > 0.60$, $P < 0.01$), whereas NKX2−1 exhibited a negative correlation with monocytes ($r = −0.50$, $P < 0.05$). Monocytes serve as precursors to macrophages [68], and the negative correlation between NKX2−1 and monocytes may reflect the impaired differentiation of monocytes into antifibrotic macrophage subpopulations in SSc-ILD. The activated dendritic cells and macrophages synergistically initiate adaptive immunity. Their positive correlation with ELF3/ARG2 indicates that these key genes may regulate dendritic cell–macrophage crosstalk, amplifying inflammatory and fibrotic signals. Notably, the SSc-ILD group exhibited higher abundances of M0 macrophages and plasma cells, which is consistent with the disease's characteristic shift of macrophages toward proinflammatory and profibrotic phenotypes. This finding further indicated that key genes exert their effects by regulating the interactions between macrophages and immune cells.

Drug target docking analysis revealed that ARG2 exhibits favorable docking binding energies with SKA-111 and cyclophosphamide, whereas ELF3 exhibits favorable docking binding energies with voruciclib and cyclophosphamide. SKA-111(5-methylnaphtho[1,2-d]thiazol-2-amine) is a calcium-activated potassium channel (KCa) activator that selectively targets the KCa3.1 channel [63]{Shim, 2019 #241}. It serves as a research tool for exploring the potential therapeutic application of KCa channels in neurological and cardiovascular diseases. Voruciclib is an oral cyclin-dependent kinase (CDK) inhibitor that inhibits CDK activity, thereby blocking cell cycle progression and transcriptional regulation to suppress tumor cell proliferation. At present, no published studies reporting on the relationship between SKA-111 and voruciclib in pulmonary fibrosis or autoimmune diseases have been found. Cyclophosphamide is an alkylating agent immunosuppressant that is currently and widely used in the treatment of SSc, particularly in patients with ILD and pulmonary arterial hypertension [63]. Cyclophosphamide exhibits favorable docking with the genes ARG2 and ELF3, indicating its potential role in regulating immune or fibrotic processes by targeting these two genes, thereby potentially influencing the pathological mechanism of SSc-ILD.

In this study, the key genes ARG2, ELF3, and NKX2–1 were found to be related to macrophages in SSc-ILD disease, which may affect signal transduction and further affect the occurrence of SSc-ILD. The results of this study establish a foundation for comprehensively understanding the underlying mechanism of SSc-ILD and enhancing clinical diagnosis and therapeutic strategies. However, this study has certain limitations. First, the conclusions are primarily based on bioinformatic analysis of public databases, lacking direct validation of gene functions and molecular mechanisms through in vitro or in vivo experiments. Second, the limited sample size of single-cell and transcriptomic data may affect the generalizability and statistical power of the results. In addition, macrophages exhibit high heterogeneity, and this study did not delve into the specific roles of key genes within distinct functional subpopulations (e.g., proinflammatory or profibrotic subtypes). Finally, although computational predictions indicate promising binding activities between potential therapeutic drugs (e.g., SKA-111 and voruciclib) and key genes, their actual efficacy, specificity, and safety in SSc-ILD models require experimental validation. Thus, future validation requires larger, multicenter independent clinical cohorts. Studies should

integrate CRISPR gene editing, animal models, and multi-omics integration strategies to elucidate the upstream and downstream regulatory mechanisms of these key genes at the experimental level. Concurrently, expanding single-cell multi-omics analyses targeting macrophage heterogeneity and conducting preclinical functional validation of candidate drugs will advance their clinical translation as novel biomarkers or therapeutic targets.

## Supporting information

**S1 Fig. Quality control of single-cell data (the violin plots of each sample after quality control, where nFeature-RNA represents the number of genes in the cell, nCunt-RNA represents UMI in the cell, and percent.mt represents the percentage of mitochondrial genes).**
(TIF)

**S2 Fig. (a) Fibrotic macrophage subgroups.** The upper figure shows that macrophages are classified into high-fibrosis and low-fibrosis groups based on the AUCell score. The higher the score, the higher the degree of fibrosis, and the color is redder. The lower figure divides macrophages into two groups: red represents high-fibrosis macrophages, and blue represents low-fibrosis macrophages. (b) The corresponding situations of different clusters. The left image corresponds to the macrophage subsets, and the right one corresponds to the high- and low-fibrosis macrophages. (c) The corresponding expression of ARG2 in the high- and low-fibrosis macrophage subsets. The redder the color, the higher the expression level of the corresponding gene. (d) The corresponding expression of ELF3 in the high- and low-fibrosis macrophage sub-sets. (e) The corresponding expression of NKX2−1 in the high- and low-fibrosis macrophage subsets.
(TIF)

**S3 Fig. (a) Scale-free index and average connectivity graph for soft thresholding.** The network with a soft threshold value of 3 best conforms to the distribution state of scale-free networks. (b) Gene clustering tree diagram based on topo-logical overlap dissimilarity, with different colors representing different gene modules and gray representing unclassified modules, resulting in eight co-expressed module colors (excluding the gray module to which unclassified genes belong).
(TIF)

**S4 Fig. (a) Chromosomal localization of key genes.** (b) Functional similarity analysis of key genes. Purple represents NKX2−1; orange represents ELF3; green represents ARG2, and the horizontal axis represents the GO functional similarity score.
(TIF)

**S5 Fig. Infiltration of immune cells in the SSc-ILD group and control group.** Red represents the SSc group, and blue represents the normal group.
(TIF)

**S6 Fig. Violin plots of known macrophage functional markers.** Blue represents cluster 5, and red represents cluster 3.
(TIF)

**S7 Fig. Correlation analysis of phagocyte function scores and key genes.** (a) Box plot of macrophage function enrichment. Different colors represent different groups; blue represents the normal group, and orange represents the disease group. (b) Correlation analysis of macrophage function and key genes. The different groups are represented by blue (negative correlation) and red (positive correlation). Asterisks indicate significance.
(TIF)

**S1 Table. GO analysis of candidate genes.**
(XLSX)

**S2 Table. KEGG analysis of candidate genes.**
(XLSX)

**S1 File. Original code.**
(ZIP)

## Author contributions

**Conceptualization:** Ting Zhao, Fu-an Lin.

**Data curation:** Ting Zhao, Yulin Wang, Fu-an Lin.

**Formal analysis:** Ting Zhao, Fu-an Lin.

**Funding acquisition:** Ting Zhao, Fu-an Lin.

**Investigation:** Ting Zhao, Yulin Wang.

**Methodology:** Ting Zhao.

**Project administration:** Ting Zhao, Fu-an Lin.

**Resources:** Ting Zhao, Fu-an Lin.

**Software:** Ting Zhao, Yulin Wang.

**Supervision:** Ting Zhao.

**Validation:** Ting Zhao, Yulin Wang, Fu-an Lin.

**Visualization:** Ting Zhao, Fu-an Lin.

**Writing – original draft:** Ting Zhao.

**Writing – review & editing:** Fu-an Lin.

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
