## [Decision Letter · Decision Letter 0]

2 Dec 2025

Dear Dr. Lin,

We look forward to receiving your revised manuscript.

Kind regards,

Eric A. Shelden, Ph.D.

Academic Editor

PLOS ONE

Journal Requirements:

“This research was funded by Zhangzhou Affiliated Hospital of Fujian Medical University Doctoral Studio, Grant number: PDA202207.”

“The research reported in this project was generously supported by [Zhangzhou Affiliated Hospital of Fujian Medical University Doctoral Studio] under grant agreement number [PDA202207].”

“This research was funded by Zhangzhou Affiliated Hospital of Fujian Medical University Doctoral Studio, Grant number: PDA202207.”

Reviewers' comments:

Reviewer's Responses to Questions

**Comments to the Author**

1. Is the manuscript technically sound, and do the data support the conclusions?

Reviewer #1: Yes

Reviewer #2: Yes

2. Has the statistical analysis been performed appropriately and rigorously?

Reviewer #1: Yes

Reviewer #2: I Don't Know

3. Have the authors made all data underlying the findings in their manuscript fully available?

Reviewer #1: Yes

Reviewer #2: Yes

4. Is the manuscript presented in an intelligible fashion and written in standard English?

Reviewer #1: Yes

Reviewer #2: No

Reviewer #1: Further detailed comments have been uploaded.

This manuscript by Lin et al. describes the crucial role of macrophages in SSc-ILD. Using computational methods and a comprehensive array of bioinformatical analyses it identified 3 key genes contributing to inflammation driven fibrosis in SSc-ILD, namely ARG2, NKX2-1 and ELF3.

Overall, a myriad of computational methods has been employed. These analytical methods are well described in the methods section. However, to make this work accessible to the non-bioinformatic audience, it would help to remind the reader of the aim of a particular analysis in the results section.

The results section needs to be re-written, ensuing that figure legends and the main text contain the same information and that each figure with its legends can stand alone.

Having identified the three key regulatory genes in SSc-ILD, the section on the drug identifications appears rather short and with less detail. This is, however, one of the most interesting results.

Apart from the detailed requested corrections outlines below, my main criticism of the work is:

1. The number of images needs to be reduced to those more informative. Others should be included in the supplementary data.

2. Having identified the interaction of macrophages with alveolar type 2 cells in figure 2 this aspect in SSc-ILD development should have been further investigated.

3. Given the vast number of analyses, the limitations mentioned appear rather small.

4. Additionally, as fibrosis is the measurable outcome leading to SSc-ILD, the interaction of macrophages with fibroblasts should have been mentioned (and further interrogated).

5. This analysis should have been performed comparing SSc and SSc-ILD to identify which cell type / gene panel drives SSc-ILD in patients originally diagnosed with SSc (which is more manageable and has a better prognosis that the diagnosis of SSc-ILD). Please comment on this.

6. The lack of biological confirmation (although several studies are quoted suggesting the importance of the identified genes in the progression of fibrosis. (E.g., Knockout of the ARG2 gene can reduce age-related pulmonary fibrosis).

Reviewer #2: The authors have carried out an important and complex evaluation of public single-cell transcriptomic datasets, which allowed them to identify genes associated with monocyte–macrophage function in SSc patients affected by ILD. Particularly noteworthy is the section linking these findings to potential drug targets, which in my view deserves greater emphasis, especially within the “Discussion” section.

Although this is a highly valuable study from a biostatistical perspective, several key aspects require clarification.

Major Comments

1. While the authors performed a significant biostatistical analysis using three public datasets and devoted much of the “Introduction” section to the role of macrophages in pulmonary fibrosis, the discussion appears lengthy and insufficiently focused on possible correlations and links between macrophages and differentially expressed genes, or on the results emerging from their analyses. In my opinion, this makes it difficult to fully appreciate the true value of the work. The discussion should therefore be substantially revised to highlight and interpret the results that are truly important—particularly those that may be useful for future translational and functional studies on macrophages (which, given the introduction, should be the primary cellular subset of interest), the pathways and signaling mechanisms involved, and the connections with pharmacological interventions.

Indeed, the “Introduction” highlights the importance of macrophages, the results highlight distinct macrophage subsets (SPP1hi and FABP4hi), as well as differentially expressed genes and gene modules between SSc-ILD patients and healthy controls, along with differences in cell types. Yet, the discussion lacks a clear narrative thread that ties all these aspects together.

The authors might consider streamlining the work, perhaps presenting fewer results, but ones that are more directly useful for understanding the study’s purpose and implications.

2. Although the authors refer to three major public datasets, it is unclear whether the reported findings derive from the analysis of all three datasets or primarily from GSE122960, which is the only dataset explicitly mentioned. This is a crucial point that must be clarified much more thoroughly.

3. The results indicate the presence of additional macrophage subgroups with varying abundance, yet these are neither reported nor discussed. Could the authors explain why this description was omitted?

4. The authors initially identified 50 candidate genes, but only 20 were ultimately evaluated. How and why was this reduction made? Are these genes specifically related to the disease? This discrepancy needs to be clarified.

5. In the “Introduction,” the authors describe the importance of macrophages in pulmonary fibrosis and SSc pathology. However, the literature cited should be updated, as several recent reviews have been published—for example:

o Cutolo M. et al., Nature Reviews Rheumatology, 2025 Sep; 21(9):546–565.

o Campitiello R. et al., Autoimmunity Reviews, 2024 Oct; 23(10):103637. doi: 10.1016/j.autrev.2024.103637.

6. In the “Results” section (lines 281 and 283), the authors write: “a relatively strong correlation.” Could they clarify what they mean by this? Is the correlation statistically significant or not?

Minor comments.

Please include the full-leigh name of molecules when cited for the first time in the manuscript (examples: TGFbeta1 and IL13 in the “Introduction” section).

A full revision of the English style is needed.

**Do you want your identity to be public for this peer review?** For information about this choice, including consent withdrawal, please see our Privacy Policy

Reviewer #1: **Yes:** Dr Bettina C Schock

Reviewer #2: No

---

## [Author Response · Author response to Decision Letter 1]

14 Jan 2026

Point-by-point response to the reviewer comments.

Dear Reviewers,

Thank you for your thoughtful suggestions and insights, which have greatly improved the manuscript. We look forward to working with you to bring this manuscript closer to publication in “PLOS ONE”.

The manuscript has been carefully rechecked, and all necessary changes have been made in accordance with your recommendations. Responses to all comments have been prepared and are attached below. We have made every effort to address the issues you raised, and we hope that the revised manuscript is now suitable for publication in “PLOS ONE”. If you have any remaining questions regarding this paper, please do not hesitate to contact us.

Sincerely,

Fu-an Lin

Journal Requirements:

Re Dear Editor, Thank you for your thoughtful reminder and professional oversight. I have promptly addressed your guidance regarding the manuscript's compliance with PLOS ONE style requirements and file naming conventions. Prior to this submission, I had already carefully formatted the manuscript in strict accordance with the journal’s official guidelines, ensuring that every detail aligns with its standards. If any areas for further improvement are identified, I will make the necessary adjustments promptly. Should additional coordination or supplementary explanations be required in the future, I will respond in a timely manner. Once again, I sincerely appreciate your patient guidance and thoughtful assistance. I wish you all the best in your work.

Re Dear Editor, Thank you for your professional guidance and detailed instructions. We apologize that, due to local network issues, we were unable to upload our original data code to the public repository. To ensure full transparency and support the reproducibility of the manuscript's findings, we have now submitted the original code and complete data as supplementary material to the journal. This approach aligns with best practices for research reproducibility and reusability, ensuring that all author-generated code is freely accessible with clear and traceable documentation. Should any further supplementation or adjustments be required, we will promptly cooperate to make the necessary changes. Once again, we sincerely appreciate your patient guidance and meticulous oversight. We wish you all the best in your work.

“This research was funded by Zhangzhou Affiliated Hospital of Fujian Medical University Doctoral Studio, Grant number: PDA202207.”

Re Dear Editor, Thank you for your timely reminder and detailed guidance. This research was indeed funded by the Doctoral Student Research Fund of Zhangzhou Affiliated Hospital, Fujian Medical University (Project No.: PDA202207). The funding agency did not participate in the study design, data collection and analysis, publication decisions, or manuscript writing. We will strictly comply with journal requirements by including this statement regarding the funding source’s role in the cover letter, ensuring the accuracy and compliance of all relevant information, as reflected in the attached cover letter file. We sincerely appreciate your professional assistance in facilitating the smooth progress of this manuscript.

“The research reported in this project was generously supported by [Zhangzhou Affiliated Hospital of Fujian Medical University Doctoral Studio] under grant agreement number [PDA202207].”

“This research was funded by Zhangzhou Affiliated Hospital of Fujian Medical University Doctoral Studio, Grant number: PDA202207.”

Re Dear Editor, Thank you for your timely reminder and detailed guidance. This research was funded by the Doctoral Student Research Fund of Zhangzhou Affiliated Hospital, Fujian Medical University (Project No.: PDA202207). The funding agency had no role in the study design, data collection and analysis, publication decisions, or manuscript writing. In accordance with journal requirements, we have included this statement regarding the funding source’s role in the cover letter to ensure the accuracy and compliance of the relevant information, as reflected in the attached file. We sincerely appreciate your professional assistance in facilitating the smooth progress of this manuscript.

Re Thank you for your message. We extend our gratitude to the reviewers and editors for their valuable suggestions and guidance. As recommended by the reviewers, we have added two additional references, 2 and 12. After evaluation, these references are deemed to hold significant scientific importance.

Reviewers' comments:

Reviewer's Responses to Questions

Comments to the Author

1. Is the manuscript technically sound, and do the data support the conclusions?

Reviewer #1: Yes

Reviewer #2: Yes

2. Has the statistical analysis been performed appropriately and rigorously?

Reviewer #1: Yes

Reviewer #2: I Don't Know

3. Have the authors made all data underlying the findings in their manuscript fully available?

The PLOS Data policy requires authors to make all data underlying the findings described in their manuscript fully available without restriction, with rare exception (please refer to the Data Availability Statement in the manuscript PDF file). The data should be provided as part of the manuscript or its supporting information, or deposited to a public repository. For example, in addition to summary statistics, the data points behind means, medians and variance measures should be available. If there are restrictions on publicly sharing data, e.g. participant privacy or use of data from a third party, those must be specified.

Reviewer #1: Yes

Reviewer #2: Yes

4. Is the manuscript presented in an intelligible fashion and written in standard English?

Reviewer #1: Yes

Reviewer #2: No

Re Dear Reviewer, We sincerely appreciate your thoughtful feedback and professional guidance. In response to your concerns regarding formatting and grammatical standards, we have submitted the manuscript to a specialized academic editing service for comprehensive refinement. Key revisions address grammatical errors, formatting inconsistencies, and ambiguous phrasing, ensuring that the text adheres to English academic writing conventions, maintains logical coherence, and conveys meaning clearly. We are grateful once again for your attentive guidance, which has significantly improved the manuscript’s overall presentation.

5. Review Comments to the Author

Reviewer #1: Further detailed comments have been uploaded.

1.This manuscript by Lin et al. describes the crucial role of macrophages in SSc-ILD. Using computational methods and a comprehensive array of bioinformatical analyses it identified 3 key genes contributing to inflammation driven fibrosis in SSc-ILD, namely ARG2, NKX2-1 and ELF3.

Overall, a myriad of computational methods has been employed. These analytical methods are well described in the methods section. However, to make this work accessible to the non-bioinformatic audience, it would help to remind the reader of the aim of a particular analysis in the results section.

The results section needs to be re-written, ensuing that figure legends and the main text contain the same information and that each figure with its legends can stand alone.

Having identified the three key regulatory genes in SSc-ILD, the section on the drug identifications appears rather short and with less detail. This is, however, one of the most interesting results.

Apart from the detailed requested corrections outlines below, my main criticism of the work is:

Re Dear Reviewer, Thank you for taking the time to conduct a thorough and rigorous review of this study and for providing highly constructive suggestions for revision. Your comments have accurately identified key areas for improvement regarding the paper’s readability, completeness, and presentation of value, offering invaluable guidance for enhancing the academic quality of our research. We have carefully reviewed each of your points and fully agree with your assessment. Regarding the three suggestions you raised, we will implement the following revisions:

Clarifying the Objectives of Supplementary Analyses: We fully acknowledge your concerns regarding the specific objectives of the supplementary analyses. To improve comprehension for readers without a bioinformatics background, we will clearly explain the core purpose of each computational analysis in the corresponding sections of the Results chapter. For instance, we will explicitly state the roles of different bioinformatics methods in identifying key regulatory genes for SSc-ILD and elucidating how inflammation drives fibrosis. Using concise and accessible language, we will create logical connections between methods and results, enabling readers from diverse fields to grasp the research approach smoothly (see the Results chapter for details).

Rewriting the Results Section and Optimizing Figure Legends: We will conduct a comprehensive review of the Results chapter. First, we will cross-reference each sentence in the main text with the corresponding figure captions to ensure strict consistency and mutual corroboration. Second, we will enhance the self-contained nature of each figure and its legend by adding necessary supplementary explanations, allowing each figure to fully convey its core findings independently. These revisions will substantially improve the paper’s standardization, readability, and overall clarity (see the Results chapter and figure captions).

Regarding Enhancements to the Drug Identification Section: We fully concur with your assessment that “the drug identification section represents one of the most valuable findings of this study” and appreciate your suggestion to present this content more fully. We have already incorporated discussion of this key discovery and have now refined and supplemented the core content. In the Discussion, we further elucidate the translational medical significance of these findings: Although SKA-111 (a KCa3.1 channel agonist) and voruciclib have not previously been reported in pulmonary fibrosis, our study is the first to suggest, via computational modeling, their potential as novel therapeutic targets for SSc-ILD. Additionally, the strong binding of cyclophosphamide to ARG2 and ELF3 provides a potential molecular mechanism explaining its known partial efficacy in SSc-ILD, likely by modulating immune or fibrotic pathways mediated through these genes (see lines 801-817 in the manuscript). Once again, we sincerely thank you for your patient guidance and expert insights. Your suggestions have greatly helped us clarify the manuscript’s optimization potential.

2.The number of images needs to be reduced to those more informative. Others should be included in the supplementary data.

Re Dear Reviewer, Thank you very much for your valuable suggestions. Your guidance on streamlining the main text figures and highlighting core information is highly appreciated. We fully agree that reducing redundant figures will clarify the paper’s focus and enhance readability. We will carefully review and refine all figures, retaining only those that directly support key research findings and provide essential information within the main text. All other figures that are of reference value but not essential will be moved to the supplementary data section, ensuring clear logic and focused emphasis in the main text. Once again, we sincerely appreciate your meticulous review and professional guidance.

3.Having identified the interaction of macrophages with alveolar type 2 cells in figure 2 this aspect in SSc-ILD development should have been further investigated.

Re Dear Reviewer, Thank you very much for your insightful and constructive suggestions. You have accurately identified a key area for further exploration in the study of SSc-ILD pathogenesis, the interaction between macrophages and alveolar type II epithelial (AT2) cells. This perspective addresses a critical extension of our research and provides invaluable guidance for enriching the scientific depth and rigor of the paper. We fully agree with and highly value your assessment. To address this direction, we will supplement and deepen the content from multiple dimensions: First, We will elaborate in detail how the LGALS9–CD44 ligand–receptor pair identified in Figure 2 specifically regulates AT2 cell functions, such as proliferation, differentiation, apoptosis, and repair capacity, through mediated signaling pathways, clarifying the molecular mechanisms underlying the inflammatory-to-fibrotic transition in SSc-ILD. Second, Integrating existing evidence, we will expand on the regulatory role of key cytokines secreted by monocyte-derived pro-fibrotic alveolar macrophages (MoAM) in AT2 cell dysfunction, and explain how M1/M2 polarization imbalance in mixed-phenotype macrophages exacerbates alveolar epithelial injury by disrupting macrophage–AT2 cell communication. Third, We will further lin

---

## [Decision Letter · Decision Letter 1]

17 Feb 2026

Identification of key genes related to macrophages in systemic sclerosis-associated interstitial lung disease based on single-cell and bulk transcriptomics data

PONE-D-25-54894R1

Dear Dr. Lin,

We’re pleased to inform you that your manuscript has been judged scientifically suitable for publication and will be formally accepted for publication once it meets all outstanding technical requirements.

Kind regards,

Eric A. Shelden, Ph.D.

Academic Editor

PLOS One

Additional Editor Comments (optional):

Reviewers' comments:

Reviewer's Responses to Questions

**Comments to the Author**

Reviewer #2: All comments have been addressed

2. Is the manuscript technically sound, and do the data support the conclusions?

Reviewer #2: Yes

3. Has the statistical analysis been performed appropriately and rigorously?

Reviewer #2: I Don't Know

4. Have the authors made all data underlying the findings in their manuscript fully available?

Reviewer #2: Yes

5. Is the manuscript presented in an intelligible fashion and written in standard English?

Reviewer #2: Yes

Reviewer #2: I thank the authors for addressing the explanation to the single comments and for improving the manuscript.

**Do you want your identity to be public for this peer review?** For information about this choice, including consent withdrawal, please see our Privacy Policy

Reviewer #2: No

---

## [Editor Report · Acceptance letter]

PONE-D-25-54894R1

PLOS One

Dear Dr. Lin,

I'm pleased to inform you that your manuscript has been deemed suitable for publication in PLOS One. Congratulations! Your manuscript is now being handed over to our production team.

Kind regards,

on behalf of

Dr. Eric A. Shelden

Academic Editor

PLOS One